# Bile acids drive the newborn's gut microbiota maturation

N. van Best [1,2], U. Rolle-Kampczyk [3], F. G. Schaap [4,5], M. Basic[6], S. W. M. Olde Damink [4,5], A. Bleich [6], P. H. M. Savelkoul[2], M. von Bergen[3,7], J. Penders [2,8,9 ✉] & M. W. Hornef [1,9 ✉]

Following birth, the neonatal intestine is exposed to maternal and environmental bacteria that successively form a dense and highly dynamic intestinal microbiota. Whereas the effect of exogenous factors has been extensively investigated, endogenous, host-mediated mechanisms have remained largely unexplored. Concomitantly with microbial colonization, the liver undergoes functional transition from a hematopoietic organ to a central organ of metabolic regulation and immune surveillance. The aim of the present study was to analyze the influence of the developing hepatic function and liver metabolism on the early intestinal microbiota. Here, we report on the characterization of the colonization dynamics and liver metabolism in the murine gastrointestinal tract ($n = 6$–10 per age group) using metabolomic and microbial profiling in combination with multivariate analysis. We observed major age-dependent microbial and metabolic changes and identified bile acids as potent drivers of the early intestinal microbiota maturation. Consistently, oral administration of tauro-cholic acid or β-tauro-murocholic acid to newborn mice ($n = 7$–14 per group) accelerated postnatal microbiota maturation.

[1] Institute of Medical Microbiology, RWTH University Hospital Aachen, RWTH University, Aachen, Germany. [2] Department of Medical Microbiology, School of Nutrition and Translational Research in Metabolism (NUTRIM), Maastricht University, Maastricht, The Netherlands. [3] Department of Molecular Systems Biology, UFZ-Helmholtz Centre for Environmental Research, Leipzig, Germany. [4] Department of General Surgery, NUTRIM, Maastricht University, Maastricht, The Netherlands. [5] Department of General, Visceral and Transplantation Surgery, RWTH University Hospital Aachen, Aachen, Germany. [6] Institute for Laboratory Animal Science, Hannover Medical School, Hannover, Germany. [7] Institute of Biochemistry, University of Leipzig, Leipzig, Germany. [8] School of Public Health and Primary Care, Maastricht University, Maastricht, The Netherlands. [9] These authors contributed equally: J. Penders, M. W. Hornef. ✉email: j.penders@maastrichtuniversity.nl; mhornef@ukaachen.de

All mammalian host body surfaces and in particular the intestinal tract are colonized by a variety of microorganisms. The intestinal microbiota of the healthy host represents a dense and competitive microbial ecosystem and provides a core set of metabolic activities and immunostimulatory molecules, which together significantly contribute to the host's metabolism, tissue development and immune maturation and protect from infection with enteropathogenic microorganisms[1]. Perturbations of the human intestinal microbiota composition have been associated with highly prevalent metabolic and immune-mediated diseases. Therefore, a detailed understanding of the mechanisms that establish and maintain a beneficial microbiota composition in the healthy host is fundamental. In particularly, this relates to the immediate postnatal and early infant period during which initial bacterial colonization accompanies dynamic fluctuations in the microbiota composition[2–5]. Notably, the early postnatal colonization process is of critical importance for the long-term microbiota composition[6], and immune maturation and thereby the susceptibility to highly prevalent non-communicable diseases[7]. Microbiota alterations during this time window have been associated with adverse consequences and an enhanced susceptibility to disease[8,9].

The fetal mucosal surfaces are sterile[10]. With birth, exposure to the maternal vaginal, fecal and skin microbiota establishes an early set of bacteria. With time, additional bacteria are incorporated generating an increasingly diverse bacterial ecosystem with significant resilience to exogenous perturbations[11]. The process of intestinal bacterial colonization after birth has been intensively studied during the last years and exogenous factors which significantly impact the early microbiota composition have been identified[3,5,12–15]. However, exogenous factors can explain only a limited amount of the inter-individual variation in microbiota composition[5,16].

Few studies have attempted to characterize host-mediated endogenous mechanisms such as for example genetic factors that shape the microbial ecosystem early after birth[17–19]. The identification and characterization of such endogenous mechanisms is expected to provide insight in the functional relevance of particular aspects of the microbiota since they result from a long process of host-microbial coevolution[20]. In this study, we aim to unravel the impact of the developing host hepatic metabolism on the murine intestinal microbiota composition. To characterize the intestinal microbiota maturation, we first profile the microbial composition of the small and large intestine obtained from animals within the same litter during the immediate postnatal period, early infancy, and weaning until adulthood. Subsequently, we perform a global metabolic screen in the liver tissue of the same animals to provide information on the site-specific metabolites. Multi-omics analysis identifies specific bile acids associated with maturation of the early intestinal microbiota and subsequent interventional studies functionally confirm and specify the role of specific bile acids as important drivers of microbiota development.

## Results

**Postnatal kinetic of the intestinal microbiota composition.** To investigate the postnatal gut microbial development, we obtained the complete small intestinal and colonic tissue from mice aged 0, 6, 12, 18, and 24 h ($n = 10$/age group) as well as 1, 7, 14, 28, and 56 days ($n = 6$/age group) after birth to cover the early developmental stages until adulthood. To avoid litter effects, tissues for all time points in a given kinetic were taken consecutively from animals from one litter (Fig. 1a and Supplementary Fig. 1a—schematic overview of study design). Amplicon sequencing of the 16S rRNA V4 gene region of these samples generated in total 15,692,587 high-quality sequences with a median of 87,227 [IQR

77,784] reads per sample. All sequences were subsequently clustered based on de novo assembly into 478 operational taxonomic units (OTUs) that were taxonomically classified to the genus level.

No substantial changes in the general community structure were observed during the first 24 h (Supplementary Fig. 1b). The bacterial richness and diversity decreased significantly in both small intestine and colon during the first 24 h after birth (Supplementary Fig. 1c and d, Supplementary Fig. 2a). In line with the reduced diversity, the relative abundance of almost all genera decreased during this early time window in both the small intestine and colon. Only the abundance of the genera *Corynebacterium* and *Mannheimia* increased temporarily with a peak at 6–18 h and 24 h, respectively. (Fig. S1e, f). The initial decrease in richness most likely indicates the transient passage of bacterial species ingested during birth and shortly thereafter which were unable to permanently colonize the intestinal tract (e.g. *Corynebacterium*). The developmental phase thus starts as indicated by the diversification from 24 h onwards (Fig. 1b–f) and we therefore decided to focus all subsequent analyses on these later time points.

Microbial richness increased from PND7 onwards in both small intestine and colon (Fig. 1b, c). While the bacterial richness was similar in both organs at birth (PND1), a significantly higher colonic richness was detected after weaning (i.e., PND28, $FDR_{adjusted}$ $p = 1.7 \times 10^{-5}$, Mann–Whitney $U$ test). Similar findings were observed for microbial diversity using the Shannon index (Supplementary Fig. 2b). The presence, origin and loss of OTUs in small intestinal and colon tissue were visualized in Sankey plots for all postnatal time points, as well as for the maternal source (Supplementary Fig. 3a, b). This analysis illustrated the contribution of OTUs from all four major phyla to the microbial ecosystem at the different time points after birth. Notably, it revealed that a significant proportion of Proteobacteria originating from an environmental (non-maternal) source (78.2% of colonic and 73.2% of small intestinal OTUs) contributed to the microbiota early after birth (PND1). In contrast, the majority of OTUs from the Actinobacteria, Bacteroidetes and Firmicutes phylum persisted throughout the examined time window and were shared with the mother (53.3–94.2% and 45.0–86.5% for the colon and small intestine, respectively). The early (PND1) Proteobacteria contribution was most evident in the colon, whereas the small intestine was most strongly influenced by Proteobacteria and Actinobacteria.

Clear separation of the small intestinal and colonic microbial composition according to age was detected by principal coordinate analysis (PCoA) (Fig. 1d). The general microbiota community structure of the small intestine and colon was indistinguishable at the early time points (PND 1–21) but diverged significantly after weaning leading to a site-specific intestinal microbiota. This result was supported by the Dirichlet multinomial mixtures (DMM) modeling in which small intestinal and colonic samples from PND 1 and 7 clustered together, whereas two distinct clusters dominated the small intestinal and colonic microbiota from PND 21 onwards (Supplementary Fig. 2c). The taxonomic changes confirmed the late emergence of an organ-specific microbiota. Whereas the abundance of the most prevalent genera *Mannheimia, Streptococcus, Escherichia, Staphylococcus*, and *Enterococcus* gradually decreased in small intestine and colon during the postnatal period, a relative enrichment in *Lactobacillus* abundance was observed in both organs (Fig. 1e, f). After weaning, the relative increase in lactobacilli continued in the small intestine whereas a more complex composition, with expansion of *Bacteroides* and members of the Porphyromonadaceae (a.o. *Muribaculum*) and Lachnospiraceae families, developed in the colon.

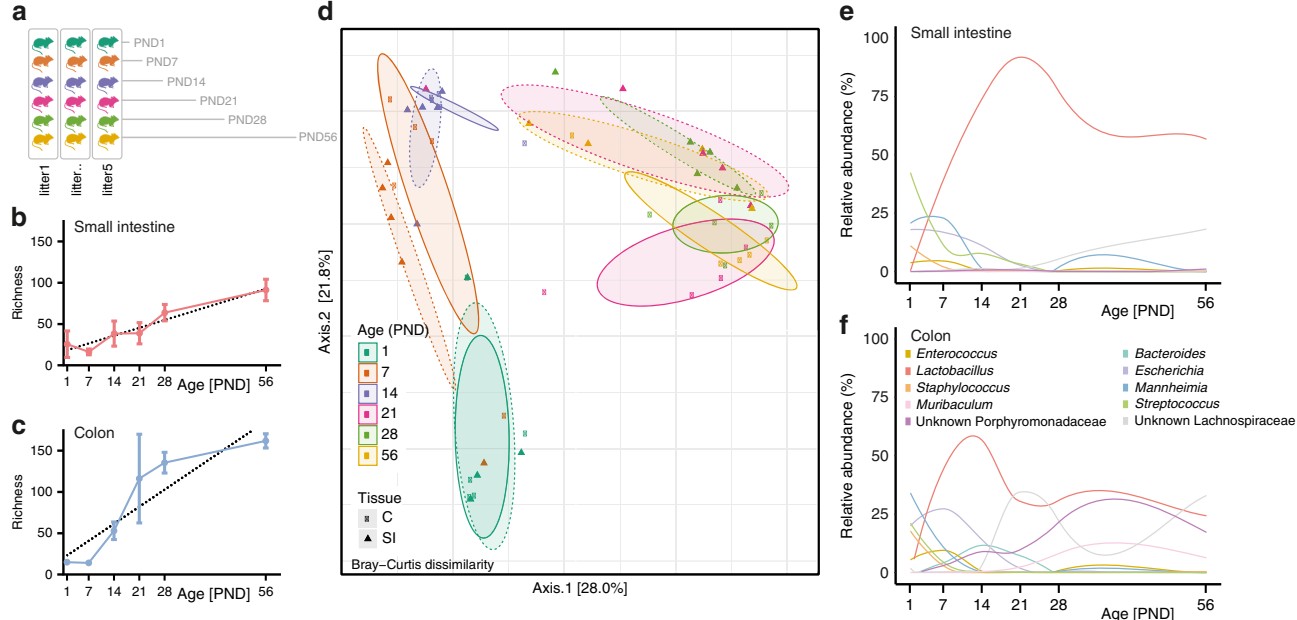

**Fig. 1 Postnatal dynamics of the intestinal microbiota composition. a** Image illustrating the study outline; one mouse from each one litter was analyzed for each individual time point (PND). **b** and **c** Richness (observed species) of the small intestinal (**b**) and colonic (**c**) microbiota exhibiting a gradual increase with age ($n = 5$ per PND for all subsequent analyses, mean and SD, $p < 0.0001$, linear regression $R^2 = 0.7702$ and $p < 0.0001$, linear regression $R^2 = 0.7281$, respectively). **d** Principal Coordinate Analysis (PCoA) based on Bray-Curtis dissimilarity indicating a gradual shift in microbial community structure along PC2 during the neonatal period PND1-PND14 and distinct structures for the small intestine and colon along PC2 at PND21-56 ($p < 0.001$, two-sided, Permanova for age and $p < 0.01$ two-sided, Permanova for tissue); Squares and solid line: colon (C); triangles and dashed line: small intestine (SI). **e**, **f** Relative abundances of the 10 most abundant genera listed in **f** between PND1-56 for the small intestine (**e**) and colon (**f**). Source data are provided as a Source Data file.

**Metabolic factors with potential influence on the microbiota.** The enzymatic and absorptive capacity of the intestine and the metabolic capacity of the liver mature significantly during the postnatal period[21–23]. The associated changes contribute to luminal substrate availability and could thereby influence the bacterial succession and overall composition[3]. To allow analysis of the same mice used for the long-term microbiota analysis presented above we investigated a panel of hepatic metabolites as a proxy for the small intestinal metabolome[24]. Major age-dependent differences were noted in particular between PND7 and PND21, 28 and 56 (Fig. 2a–d). Amino acids, biogenic amines, acylcarnitines and some glycerophospholipids showed a moderate but significant decrease between these timepoints, most likely due to the switch in diet during weaning (Fig. 2b–d), which is normally completed after day PND21. In contrast, a number of bile acids increased significantly after weaning in particular at PND7 vs. PND28, as well as PND7 vs. PND56 (Fig. 2c, d).

Constraining the ordination of all metabolites at the group level for age using Redundancy Analysis (RDA) indicated that variation in acylcarnitines, amino acids, glycerophospholipids, and sphingolipids was mainly the result of the marked differences between PND1 and all other age-groups (Supplementary Fig. 4a–c, Supplementary Fig. 5). In contrast, enterohepatic bile acids explained a major part of the age-dependent metabolic variation thereafter as depicted by RDA2 and RDA3 (Supplementary Fig. 4d, e). These age-dependent metabolic shifts were further supported using unconstrained ordination of all metabolic groups separately, illustrating the compositional changes (Supplementary Fig. 5a–i).

**Postnatal maturation of bile acid synthesis, modification, and reabsorption.** Consistently, the bile acid composition revealed a clear separation according to age along the second and third component with strong differences between birth and adulthood based on PCA (Fig. 2e). The global compositional shifts were supported by transcriptional changes of genes involved in bile acid synthesis in hepatic tissue. Transcripts for key enzymes of the classic pathway of bile acid synthesis involved in cholic acid and chenodeoxycholic acid formation such as *Cyp7a1* and *Cyp27a1* increased early (PND7–14) followed by increased *Cyp8b1* mRNA (Fig. 2f). Moreover, the key enzyme of the alternative pathway leading to chenodeoxycholic acid formation, *Cyp7b1*, was barely expressed until PND28 and unlikely to significantly contribute to synthesis of this bile acid within this time period. In contrast, the early elevation of Cyp2c70 mRNA may in particular drive increased conversion of chenodeoxycholic acid to muricholic acids, along with cholic acid major bile acid species in mice (Fig. 2g). Notably, postnatal upregulation of the key enzyme of the classical bile acid synthesis pathway *Cyp7a1* was also observed in germ-free animals. Germ-free animals are bred and raised in the absence of viable bacteria and thus this postnatal upregulation occurred in a developmental, microbiota-independent fashion (Fig. 2h).

Age-dependent changes in the concentration of bile acids were noted within the time window of major alterations in the microbiota composition. Since the metabolome analysis was performed on liver tissue to reflect the situation in the small intestine, these changes throughout the postnatal time period primarily reflected the abundance of primary bile acids. Nevertheless, prediction of the metabolic capacity of the developing microbiota by means of Phylogenetic Investigation of Communities by Reconstruction (PICRUSt) using the novel integrated Mouse Gut Metagenomic Catalog (iMGMC) revealed an age-dependent enrichment of bacteria encoding for genes involved in bile acid metabolism in adults (PND56) compared to neonates (Supplementary Fig. 6). More specifically, the predicted

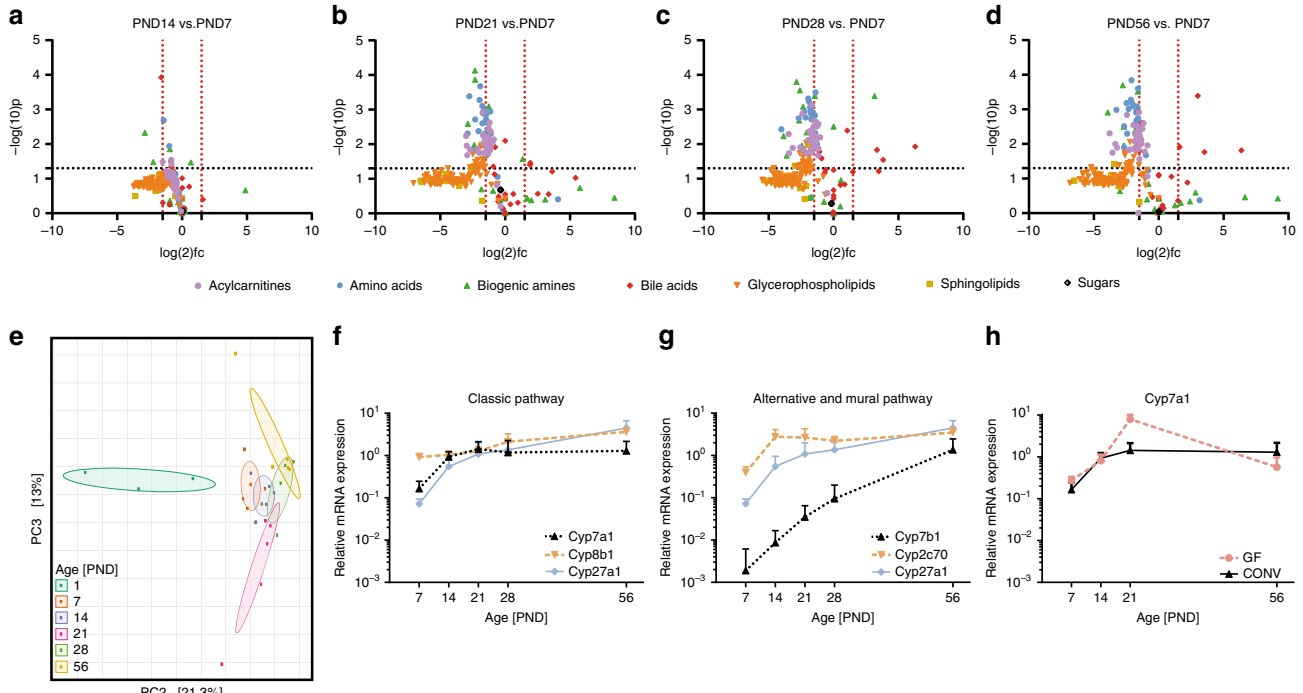

**Fig. 2 Metabolic changes during the postnatal period. a–d** Volcano plots depicting acylcarnitines, glycerophospholipids, sphingolipids, sugars, bile acids, amino acids, and biogenic amines in liver tissue at PND14 vs. PND7 (**a**), PND21 vs. PND7 (**b**), PND28 vs. PND7 (**c**), and PND56 vs. PND7 (**d**) measured by mass spectrometry ($n = 5$ per group, two-tailed independent student's t-test, fold-changes were calculated based on mean values by dividing the PNDx to PNDy concentrations and log2 the values). PND7 was selected as baseline for comparisons due to the high interindividual variation at PND1. **e** Principal Component Analysis (PCA) illustrating changes in the composition of the bile acid pool during the postnatal period. **f–g** Relative hepatic expression of the key-enzymes of (**f**) the classical (Cyp7a1, Cyp8b1, and Cyp27a1) and (**g**) the alternative and murine pathway (Cyp7b1, Cyp2c70, and Cyp27a1) ($n = 5$ per group, mean and SD). **h** Relative expression of Cyp7a1 in total liver tissue of germ-free (GF) and conventionally (CONV) raised mice ($n = 5$ per group, mean and SD). Source data are provided as a Source Data file.

abundance of genes for the key enzyme involved in bacterial bile acid metabolism, the bile salt hydrolase (BSH, KO1442), increased steadily with age reaching the highest small intestinal levels approximately at PND 21–28 (Fig. 3a). Notably, much higher abundances of BSH activity harboring bacterial taxa are observed in the large intestine. In addition, a steady increase in the expression of genes encoding proteins involved in the hepatobiliary transport of bile acids such as *Ntcp* (*Slc10A1*), *Abcb11*, and *Abcb4*, as well as proteins involved in the Fxr-Fgf15-Fgfr4 negative feedback loop regulating de novo hepatic bile acid synthesis was observed after birth until PND 56 in total liver tissue as measured by RT-PCR (Supplementary Fig. 7). Consistent with enhanced BSH activity and maturation of the hepatobiliary transport system, the hepatic concentration of bacterially modified secondary bile acids increased with age, whereas the concentration of primary bile acids decreased (Fig. 3b). Thus, the maturation of the host's bile acid biosynthetic capacity and hepatobiliary transport system in combination with an enhanced bacterial bile acid metabolism led to a marked change of the total bile acid pool as well its composition after weaning. The small intestine during the immediate postnatal period, however, was characterized by the presence of mainly conjugated, primary bile acids.

**Integrative analysis of metabolomic and microbiota data.** To investigate the potential influence of bile acid exposure on the composition of the small intestinal microbiota, we next characterized the relationship between individual bile acids and microbial OTUs using regularized canonical correlation analysis (rCCA). rCCA aims at maximizing the correlation of two data sets obtained from the same animals. The first two components

explained most of the variance with canonical correlations >0.3 (27.1% of variation in OTUs and 54.1% of variation in bile acids). Ursodeoxycholic acid (UDCA) and glycine-conjugated cholic acid (GCA) obtained the highest canonical coefficient on the first and second component, respectively (Fig. 3c, d). Taurine-conjugated α/β-muricholic acid (TMCA) and taurine-conjugated cholic acid (TCA) exhibited the strongest negative coefficient. Consistently, UDCA and GCA negatively correlated with early colonizers such as *Mannheimia* and *Streptococcus* and positively correlated with bacteria capable to metabolize bile acids such as *Enterorhabdus* and *Lactobacillus* (Fig. 3e and Supplementary Fig. 8). In contrast, TMCA showed the opposite effect exhibiting a negative correlation with members of the Coriobacteriaceae family and a positive correlation with *Mannheimia* and *Streptococcus*. Notably, these patterns are in line with the observed changes in the concentration of specific bile acids between pre-weaning and post-weaning but also early after birth and after weaning (Supplementary Fig. 9). Whereas UDCA and GCA increased in molarity towards adulthood, the concentration of TMCA and TCA decreased over time. rCCA analysis of bile acids and microbial OTUs of the colon revealed a similarly high canonical coefficient and dominant influence of TCA, MTCA and UDCA on the colonic microbiota composition (Supplementary Fig. 10). This might be due to the high concentration of conjugated bile acids during the postnatal period or a strong influence of the small intestinal microbiota on the colonic microbiota during this time window (Fig. 3b). Thus, UDCA, GCA, TMCA, and TCA are significantly associated with the compositional changes of the microbiota with age and represent good candidates of host-derived factors that drive postnatal establishment of the enteric microbiota.

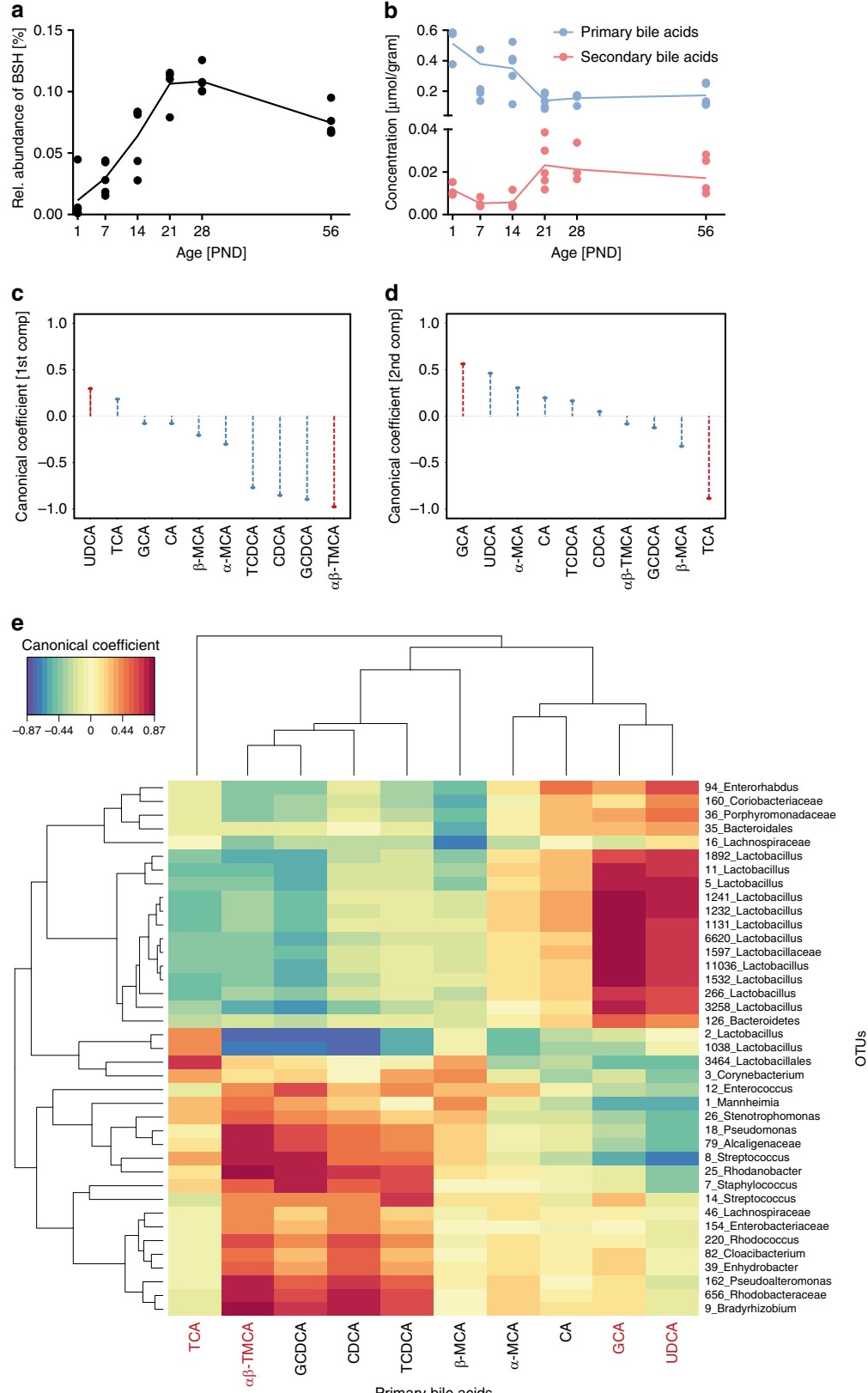

**Fig. 3 Enterohepatic cooperation in the maturation of the microbiota. a** Relative abundance of the predicted bacterial bile salt hydrolyses (KO1442) in the small intestinal microbiota ($n = 5$ per group, means connected and replicates). **b** Total concentration [μmol / gram] of primary (blue) and secondary (red) bile acids ($n = 3$ for PND1 and $n = 5$ for PND7-56, means connected and replicates). **c, d** Coefficients of the regularized canonical correlation analyses (rCCA) indicating the relationship between primary bile acids and small intestinal OTUs on the first component (**c**) and second component (**d**) (selected bile acids are highlighted in red). **e** Correlation heatmap based on the coefficients of the rCCA between hepatic primary bile acid concentrations and relative abundances of small intestinal OTUs indicated a strong positive effect (red) of GCA and UDCA on many *Lactobacillus* OTUs. Source data are provided as a Source Data file.

**Bile salt-mediated maturation of the neonatal gut microbiota.** Next, we therefore administrated the four bile acids UDCA, GCA, βTMCA, and TCA orally to neonate mice and evaluated the impact on the developing small intestinal microbiota. Neonate mice received the indicated bile acid or PBS as control by oral gavage for three consecutive days between PND7-PND9. At PND9, the site-specific small intestinal microbiota composition was determined. PCoA revealed that both TCA and βTMCA shifted the global microbiota composition towards a more adult-like microbiota composition (Fig. 4a–b). Consistently, administration of TCA and βTMCA significantly increased the richness of the small intestinal microbiota (Fig. 4c) and decreased the distance to the adult microbiota composition based on the Bray–Curtis dissimilarity analysis (Fig. 4d). In contrast, GCA and UDCA had no impact on bacterial richness (Fig. 4c) or microbiota maturity i.e., the similarity to the adult microbiota (Fig. 4d).

A bile acid-induced shift of the small intestinal microbiota composition towards a more mature phenotype was further supported by the relative abundances of the two most abundant genera in the small intestine, *Lactobacillus and Escherichia* (Fig. 4e, f). Here, inoculation of βTMCA and UDCA enhanced the abundance of *Lactobacillus*, whereas βTMCA significantly decreased the abundance of *Escherichia*. Using the *Lactobacillus: Escherichia* ratio as a proxy for microbiota maturity, βTMCA administration led to the highest ratio with a value similar to adult animals (Fig. 4g).

Interestingly, members (OTUs) of the lactobacillus genus responded differently to individual bile acids. Whereas βTMCA enhanced OTUs that were found abundantly in adults (Fig. 5a–c), UDCA influenced only OTUs with a minor contribution of the adult microbiota (Fig. 5d, e). Moreover, the annotation of these OTUs into a phylogenetic tree with their closely related *Lactobacillus* ancestors clearly indicated a genetic distinction between the 'UDCA-responders' and 'βTMCA-responders' by falling into two separated key branches (Fig. 5f). Transformation of the OTU abundance in absolute bacterial numbers confirmed that βTMCA increased the colonization density of *Lactobacillus* OTUs found abundantly in the adult microbiota (Supplementary Fig. 11e) and decreased the colonization of *Escherichia* (Supplementary Fig. 11e). Although UDCA also decreased the absolute abundance of *Escherichia* (Supplementary Fig. 11e), it increased the colonization density of *Lactobacillus* OTUs not typically found abundantly in the adult microbiota (Supplementary Fig. 11a–c). This finding may explain why UDCA does not enhance the similarity of the neonatal microbial composition to the adult microbiota (Fig. 4d). Notably, in vitro analysis of the effect of UDCA, GCA, βTMCA, and TCA on murine *Lactobacillus* isolates revealed a direct growth promoting effect of βTMCA on the 'βTMCA-responders' *L. johnsonii* (BA_OTU 4) and *L. reuteri* (BA_OTU 7 and 364) but not on the 'UDCA-responder' *L. murinus* (BA_OTU 796 and 2) consistent with our in vivo results (Supplementary Fig. 12). UDCA exerted an inhibitory effect on the murine *Escherichia* isolate both in vitro and in vivo (Supplementary Fig. 11e, Supplementary Fig. 12). In contrast, an inhibitory effect of UDCA on *Lactobacillus* isolates was observed in vitro but not in vivo (Supplementary Fig. 11 and Supplementary Fig. 12).

## Discussion

Together, our results identify bile acids as host factors driving the postnatal small intestinal microbiota composition. They are in accordance with the results of a recent large human twin study that associated metabolic genes with the intestinal bacterial colonization[18]. The postnatal increase in hepatic bile acid synthesis and maturation of the enterohepatic cycle might

therefore significantly contribute to shape the early intestinal microbiota. Alterations in the intestinal microbiota composition have previously been observed when supplementing bile acids to adult mice[25], following bile duct ligation in mice[26], as well as in human adults with cholestasis[27]. Yet, these changes have not been linked to any microbiota-driven functional benefit for the host. In contrast, our results in mice identify specific bile acids that promote characteristics of a mature and beneficial microbiota during the particularly important immediate postnatal period.

Careful microbiota profiling throughout the postnatal period revealed major age-dependent compositional changes. Consistent with previous reports, monitoring the immediate postnatal time window revealed an initial reduction of microbial richness within the first 24 h after birth[2,28,29]. The reduction in microbial diversity immediately after birth most likely illustrates the inability of many bacterial species ingested during or shortly after birth to find their niche in the neonatal intestine and thrive. Alternatively, it may reflect the perinatal ingestion and intestinal passage of bacterial DNA remnants. Although not studied in detail so far, it is consistent with the results from previous studies in animals and humans that included samples from this early time window after birth[2,30,31]. Few bacterial genera such as *Mannheimia* and *Corynebacterium* exhibited a transient increase in abundance early after birth. The underlying mechanisms are unclear but might involve changes in exposure or altering environmental conditions in the gut lumen such as oxygen concentration, changes in substrate availability and bacterial competition, as well as the induction of mucosal host responses. Global bacterial diversification and increasing richness started only thereafter and continued until after weaning in both small intestine and colon.

We analyzed the hepatic metabolome of the identical animals with the goal to reveal the site-specific spectrum of metabolites. This strategy has previously been shown to reliably reflect the metabolic situation in the small intestine[24,32]. The complexity and inter-individual variation of both the metabolome and the gut microbiota prompted us to perform an integrative analytical approach by rCCR analysis. This analysis identified bile acids as potential drivers of the microbiota composition. Concomitant with the transition from a hematopoietic tissue to a central metabolic organ, the capacity of the human liver for bile acid biosynthesis increases[23]. Our metabolomic and transcriptional analyses are consistent with the previous notion that the human neonatal liver is able to produce and secrete a significant amount of bile acids into the small intestine[33,34]. Conjugated primary bile acids dominate during the immediate postnatal period and also after weaning are present at high concentrations in the small intestine. They may well exert a significant influence particularly on the small intestine where a critical influence of the microbiota composition on the maturation of the mucosal immune system in the neonate has been demonstrated[35,36]. Continuous maturation of the microbiota composition subsequently initiates the well-described bacteria-mediated modification of bile acids as illustrated by the rise in bile salt hydrolyze (BSH) encoding genes during weaning.

Bile acids due to their amphipathic character compromise the bacterial cell wall integrity and thereby exert a potential direct antibacterial effect, consistent with our results for UDCA[37]. In vivo, they might additionally synergize with other antibacterial molecules such as bacteriocins or host-derived antimicrobial peptides present in the mammalian intestinal lumen that also target the bacterial surface[38,39]. In accordance with our results, a great species-specific variability in the bile acid tolerance has been reported, although some general phylogenetic differences exist[37,40]. In addition, specific bile acids in mice and men act as agonists or antagonists of mucosal receptor molecules that

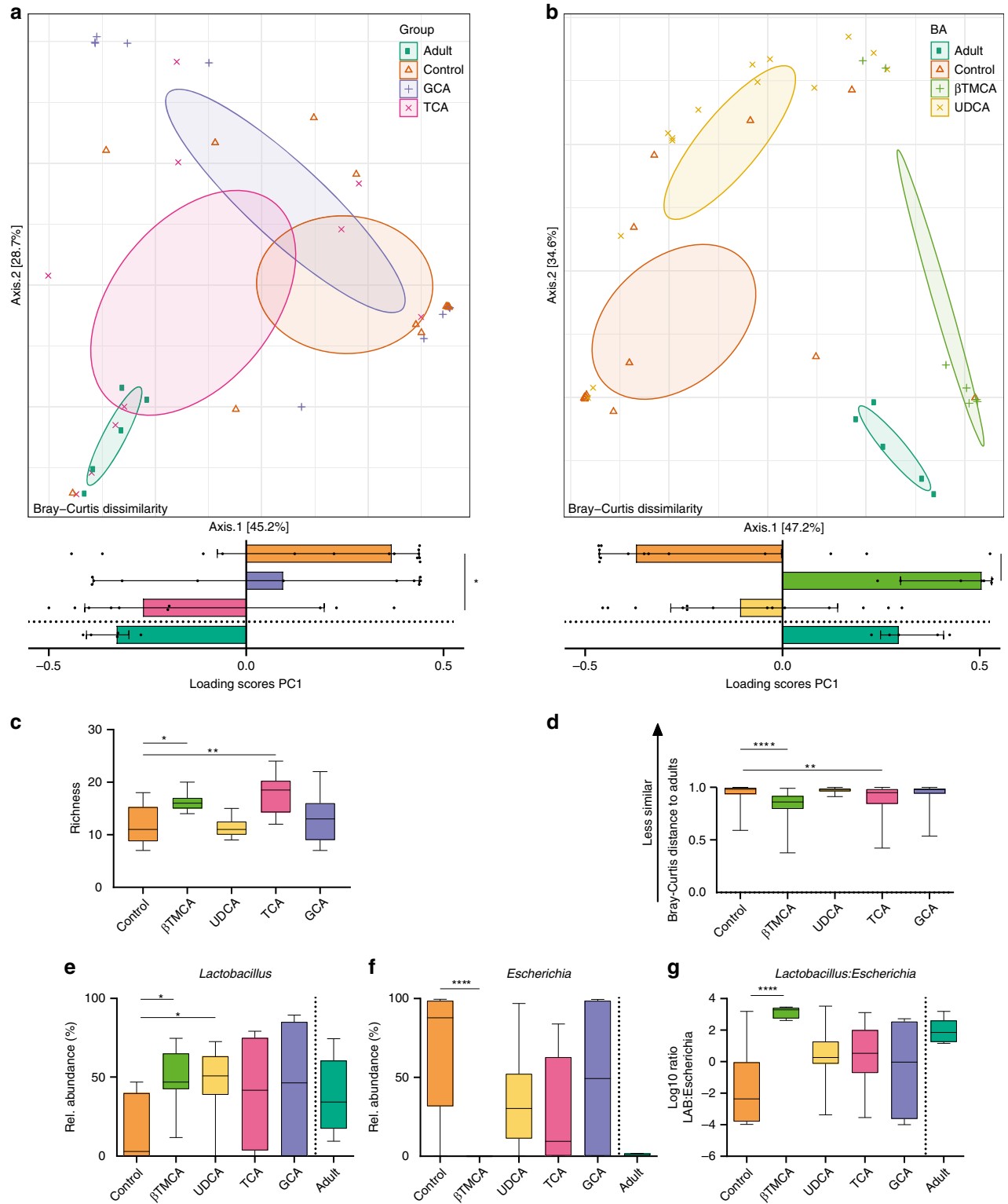

regulate the secretion of soluble mediators and influence the mucosal immune system and the global host metabolism. Induced changes may in turn indirectly influence the postnatal intestinal microbiota[41]. Specific direct or indirect effects might explain why the overall effect size of a given bile acid might not primarily depend on the concentration. Also, the described activities are expected to differentially affect bacterial taxa and thus, bile acids via a number of mechanisms could influence the composition of the intestinal microbiota.

Oral administration of two major conjugated bile acids, βTMCA and TCA confirmed their functional activity. Both taurine-conjugated bile acids enhanced the richness of the neonatal small intestinal microbiota and rendered it more similar to an adult composition. βTMCA and TCA may suppress bacteria of the neonatal microbiota thereby generating new niches for incoming bacteria and foster the abundance of members of the adult microbiota. Consistently, βTMCA lowered the abundance of *Escherichia* and enhanced the abundance of lactobacilli

**Fig. 4 Oral administration of bile acids to neonate mice accelerates microbial maturation. a, b** PCoA (upper panels) and scores of the 1st axis (lower panels) based on Bray–Curtis dissimilarity indicating shifts in microbial community structure in the small intestine of adult mice (Adult, dark green), untreated neonate mice (Control, orange), and neonate mice after oral administration of (**a**) GCA (violet) or TCA (pink) and (**b**) βTMCA (light green) or UDCA (yellow) (Kruskal–Wallis test to controls with Dunn's post-test and correction for multiple comparisons, median $+/-$ interquartile range; *$p = $ 0.0307; ***$p = 0.0001$, two-sided. $n = 14$ for UDCA and Control, $n = 11$ for GCA, $n = 10$ for TCA, $n = 7$ for βTMCA, $n = 5$ for Adult, and for subsequent analyses in this figure). **c** Richness (observed species) of the intestinal microbiota of untreated mice (Control) and mice after oral administration of GCA, TCA, βTMCA or UDCA (Kruskal-Wallis test to controls with Dunn's post-test and correction for multiple comparisons, box represents IQR with median, whiskers represent minimum and maximum values; *$p = 0.0319$; **$p = 0.0042$, two-sided). **d** Bray-Curtis distance of untreated neonate mice (Control) and neonate mice after oral administration of GCA, TCA, βTMCA, or UDCA compared to adult animals (Kruskal–Wallis test to controls with Dunn's post-test and correction for multiple comparisons, box represents IQR with median, whiskers represent minimum and maximum values; **$p = 0.0026$; ****$p < 0.0001$, two-sided). **e–g** Relative abundance of the two most abundant genera *Lactobacillus* and *Escherichia* (**e**, **f**) and ratio of the abundance of *Lactobacillus* OTUs to *Escherichia* OTUs (**g**) in the microbiota of untreated neonate mice (Control) as compared to neonate mice after oral administration of GCA, TCA, βTMCA or UDCA (Kruskal-Wallis test to controls with Dunn's post-test and correction for multiple comparisons, box represents IQR with median, whiskers represent minimum and maximum values; *$p = 0.02$; ****$p < 0.0001$, two-sided). The abundance of these genera (**e**, **f**) and the ratio of the abundance of *Lactobaccillus* OTUs to *Echerichia* OTUs (**g**) in adult mice (Adult) was added to allow visual comparison but was not included in the statistical evaluation ($n = 5$). Source data are provided as a Source Data file.

inverting the *Lactobacillus:Escherichia* ratio in an adult-like manner. Notably, bile acid administration did not influence all members of the *Lactobacillus* genus in a uniform manner. βTMCA administration in vivo enhanced the abundance *Lactobacillus* OTUs that according to available databases contain BSHs and were closely related to dominant murine *Lactobacillus* species, including *L. reuteri* (BA_OTU 7 and 364) and *L. johnsonii* (BA_OTU 4)[42]. In contrast, OTUs related to species not containing BSHs (*L. fermentum* and *L. pontis*) remained unaffected[43]. BSHs have different catalytic efficiencies and substrate preferences[44]. Some species are known to contain multiple BSHs with different substrate specificity whereas others only carry a single BSH (e.g. *L. reuteri*)[43]. The carriage of multiple BSHs by the majority of *L. johnsonii* strains might explain why TCA also enhanced the abundance of BA_OTU 4 in vivo while it did not affect the OTUs related to *L. reuteri*. Indeed, it has recently been shown that BSHs from *L. johnsonii* significantly vary in substrate specificity[45]. Consistently, βTMCA exposure in vitro strongly promoted the growth of *L. johnsonii* and, to a lesser extent, *L. reuteri*. In contrast, the growth of the BSH-negative bacteria *L. murinus* and *E. coli* remained unaffected.

Notably, the administered bile acid may exert its effect not only directly but also following bacterial modification to CA or DCA or via the induction of metabolic host changes as discussed above. In contrast to βTMCA and TCA, UDCA and GCA had no influence on richness and microbiota maturation. UDCA is highly soluble and generally considered a non-toxic bile acid with little activity on the host's bile acid receptors. GCA is a representative of the glycine-conjugated bile acids, which are common in humans but rare in mice. Substrate specificity and low glycine-deconjugating enzymatic capacity in the murine host's microbiota may therefore explain the lack of a significant effect of GCA on microbiota richness and maturation[46].

However, associations between microbial and bile acid species as observed in our multi-omics rCCA analysis did not completely match with the microbial changes induced upon oral bile acid administration. These differences may reflect the time course of bile acid exposure, the circadian luminal concentration and the dosing. In addition, the metabolic transformation of administered bile acids e.g. by deconjugation or transformation of GCA and TCA to CA and DCA may contribute. Although human neonates other than human adults were shown to produce a more taurine-dominated bile acid conjugation (similar to what is seen in mice), differences exist between the dominant bile acid species in mice and men and our findings may not completely reflect the human setting. For instance, the classic pathway is the predominant pathway for synthesis of bile acids in human liver,

whereas all pathways including the mural pathway contribute about equally to bile acid synthesis in rodents. However, ethical concerns impede the analysis of healthy neonatal liver and small intestinal tissue samples in human neonates. Human fecal samples cannot be used since they mainly contain deconjugated and secondary bile acids[34]. Likewise, the fecal bacterial composition does not reflect the small intestinal microbiota[34]. Future studies need to find ways to analyze human neonatal liver and small intestinal tissue samples and extend the analysis to rare bile acid species. Moreover, further insight in the underlying mechanisms through which bile acids influence the microbiota maturation at the various anatomical sites of the intestinal tract is needed.

Insight into the ecological factors that shape the microbiome during early life is particularly important. These factors act during the non-redundant time-window after birth that critically primes the host's immune system with life-long consequences on the susceptibility to inflammatory and immune-mediated diseases[7,13]. From the perspective of ecological theory, bile acid concentrations and species profiles could potentially promote processes such as environmental filtering or niche-based interactions. The extent to which bile acids contribute to the inter-individual variation in microbiota composition in humans should be the focus of future research.

Limitations of our study are the lack of metagenomic data and the restricted pool of metabolites analyzed in hepatic tissue. Metagenomic information would provide additional insight into the functional characteristics of the identified taxa and allow strain-level correlations with detected metabolites. Similarly, measurement of a more extended set of metabolites or even an untargeted metabolic approach could reveal other compound classes that play an important role in the development of the early postnatal microbiome. Also, the analysis of the colonic microbiota and bile acid metabolism might provide further insight. Additional factors that contribute to the microbiome development after birth might be identified by the future analysis of metabolites in other samples such as feces and blood.

In conclusion, we identify bile acids as host factors that shape the postnatal intestinal microbiota composition. The transition of the liver from a hematopoietic tissue to a central organ of metabolic regulation after birth might therefore significantly contribute to the postnatal establishment of host-microbial homeostasis. Our results might help to identify strategies to compensate for the adverse effects of necessary medical interventions such as early-life antibiotic treatment and to promote a microbial composition that is consistent with a beneficial host-microbial homeostasis.

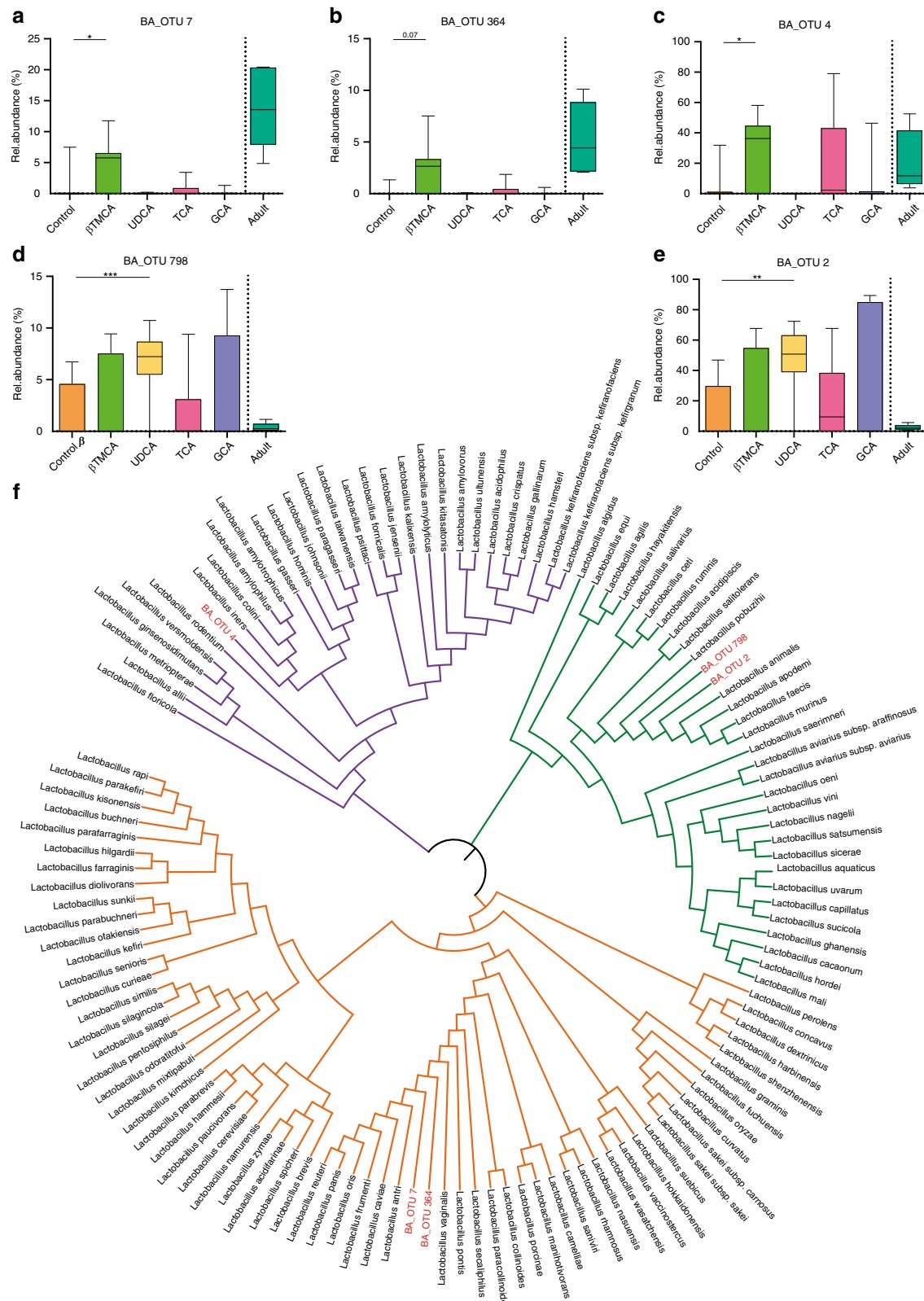

## Methods

**Ethic statement**. All animal experiments were performed in compliance with the German animal protection law (TierSchG) and approved by the local animal welfare committees, the Landesamt für Natur, Umwelt und Verbraucherschutz, North Rhine Westfalia (84–02.04.2016.A207 and 84–02.04.2015.A293). All C57BL6J wildtype mice were bred locally and held under specific pathogen-free conditions at the Institute of Laboratory Animal Science at RWTH Aachen University Hospital. The day of birth was considered day 0, i.e., animals screened at

day 1 were approximately 24 h old and verified to have ingested breast milk (abdominal milk spot). Mice were weaned at PND21.

**In vivo study design**. To monitor microbiota and host metabolic development throughout the neonatal period into adulthood, intestinal and hepatic tissues were obtained from C57BL/6 J mice in two different approaches. First, total small intestinal, colon, and liver tissues were obtained from C57BL/6 J mice aged 1, 7, 14,

**Fig. 5 Differences in the impact of bile acids on the phylogenetic lactobacillus clusters. a–e** Relative abundance of the *Lactobacillus* OTUs (**a**) BA OTU 7, (**b**) BA_OTU 364, (**c**) BA_OTU 4, (**d**) BA_OTU 798, and (**e**) BA_OTU 2 (Kruskal–Wallis test to controls with Dunn's post-test and correction for multiple comparisons, box represents IQR with median, whiskers represent minimum and maximum values; (**a**)*, p = 0,0104; (**c**)*, $p = 0,0161$; **$p = 0,0091$; ***$p = 0,0008$, two-sided, $n = 14$ for UDCA and Control, $n = 11$ for GCA, $n = 10$ for TCA, $n = 7$ for βTMCA, $n = 5$ for Adult, and for subsequent analyses in this figure) in the small intestinal microbiota of adult mice (Adult), untreated neonate mice (Control) and neonate mice after oral administration of GCA, TCA, βTMCA, or UDCA. **f** Phylogenetic tree of lactobacilli (Ezbiocloud) based on the 16S rDNA gene. The three colored branches indicate the main clusters; identified OTUs are assigned to the clusters and highlighted in red (purple, BA_OTU 4; green, BA_OTU 798 and BA_OTU 2; orange, BA_OTU 7 and BA_OTU 364). The phylogentic tree was constructed by MEGA7 version 7.0.21 (MUSCLE) for alignment and iTOL v4 for the final annotations. Source data are provided as a Source Data file.

21, 28, and 56 days ($n = 5$ per timepoint). In a separate set of experiments, similar tissues were obtained from mice aged 0, 6, 12, 18, and 24 h ($n = 10$ per timepoint). To rule out potential litter and cage effects, we obtained tissues from a single animal of one litter for a given age group and repeated this by selecting a new animal from the same litter for every other age group (Fig. 1a). Every animal was only examined once at the indicated age (PND). This means that in total 30 animals from 5 litters were used for monitoring the microbiota development between age 1 and 56 days and 46 animals from 10 litters to monitor the microbiota development within the first 24 h. For oral bile acid administration, 7–14 (UDCA and Control, $n = 14$;, GCA, $n = 11$; TCA, $n = 10$; βTMCA, $n = 7$) PND7 animals received the indicated bile acid (Sigma-Aldrich, Biomol) at a concentration of 70 μg/g body weight or PBS daily for 3 days by oral gavage (average 5 μL) with 100% succession[47]. Body weight and food/water consumption was monitored daily. To determine the effect of bile acid administration, the intestine was aseptically cut into 10–20 parts and alternately assigned to two collections for microbial profiling and assessment of metabolites. Samples from an additional group of adult 8–12 week old animals ($n = 5$) were processed in parallel to provide an adult microbiota control. Tissues were transferred into sterile micro-centrifuge tubes and stored at −80 °C before analysis. Liver tissues of 7-, 14-, 21- and 56-day-old germ-free mice were obtained from the Institute for Laboratory Animal Science at Hannover Medical School. Tissues were collected and stored in sterile tubes stored at −80 °C until further analysis.

**DNA isolation and generation of sequence data.** Total metagenomic DNA was isolated from snap frozen small intestinal and colonic tissues by repeated-bead-beating (RBB) combined with chemical lysis plus a column-based purification method[48]. Approximately 200 mg tissue was added to a 2.0 mL screw-cap tube containing 0.5 g of 0.1 mm zirconia beads (Biospec Products, Bartlesville, OK, USA) and 1 mL ASL lysis buffer from the QIAamp DNA Stool Mini Kit (Qiagen, Hilden, Germany). Samples were incubated at 15 min. at 95 °C and subsequently two successive rounds of bead-beating were employed using a FastPrep®−24 instrument (MP Biomedicals, Inc., USA) at 5.5 ms⁻¹ (3 × 1 min. for each round). To minimize physical damage of DNA in the RBB method, the lysate fraction produced from the first round of bead beating was drawn after centrifugation at full speed (~160,000 × g) for 5 min at 4 °C. A second round of bead beating was performed upon adding 300 μL fresh lysate buffer after which supernatants were pooled. To precipitate nucleic acids, 260 μL 10 M ammonium acetate was added to the lysate tubes, mixed and incubated on ice for 5 min. After centrifugation at 4 °C for 15 min. at full speed, supernatants were transferred to a new 1.5 mL tube to which an equal volume of isopropanol was added. Next, samples were incubated on ice for 30 min., centrifuged at RT for 15. min. at full speed and supernatants were removed by decanting. The nucleic acid pellet was washed with 500 μL 70% EtOH and dried under vacuum for 3 min. The nucleic acid pellet was dissolved in 100 μL TE (Tris-EDTA) buffer. Two microliter DNase-free RNase (1 mg/mL) was added and samples were incubated at 37 °C for 15 min. Next, 15 μl proteinase K and 200 μL AL buffer were added, samples were vortexed for 15 s. and incubated at 70 °C for 10 min. After adding 200 μL ethanol (96–100%) and vortexing, the lysate was transferred to QIAamp spin columns (Qiagen, Hilden, Germany). The DNA was finally purified using the QIAamp DNA Stool Mini Kit according to the manufacturer's instructions and eluated in 200 μL AE Buffer. For each DNA isolation batch, additional isolation was performed on PCR-grade water as a negative control. Generation of amplicon libraries and sequencing was performed according to a previously published protocol[49]. Briefly, the V4 region of the 16S rRNA gene was PCR amplified from each DNA sample in duplicate in 50 μL volumes containing 10 pmol of both primers (5′-GTGCCAGCMGCCGCGGTAA*-3′ [515 F] and barcoded 5′-GGACTACHVGGGTWTCTAAT*-3′ [806 R]), 5 μL Accuprime buffer II, 0.2 μL Accuprime Hifi polymerase (Thermo Fisher Scientific, Waltham, WA, USA) and 2 μL DNA. After an initial denaturation step at 94 °C for 3 min., amplification was carried out for 30 s. at 94 °C, 45 s at 50 °C and 1 min. at 72 °C. Amplification was carried out in 35 cycles for PND1-PND56; the samples with a low yield of DNA required 40 cycles (0–24 h) The PCR program ended with a final post-PCR incubation step of 10 min at 72 °C to promote complete synthesis of all PCR products. Pooled amplicons from the duplicate reactions were purified using AMPure XP purification (Agencourt, Massachusetts, USA) according to the manufacturer's instructions and subsequently quantified by Quant-iT PicoGreen dsDNA reagent kit (Invitrogen, New York, USA). Amplicons were mixed in

equimolar concentrations, to ensure equal representation of each sample, and sequenced on an Illumina MiSeq instrument using the V3 reagent kit (2 × 250 cycles). All V4 16S rDNA bacterial sequences generated in this study have been submitted to the Qiita and ENA databases under accession No. 10719 and ERP116798, respectively.

**Sequence analysis.** Data demultiplexing, length and quality filtering, pairing of reads and clustering of reads into Operational Taxonomic Units (OTUs) at 97% sequence identity was performed using the online Integrated Microbial Next Generation Sequencing (IMNGS, www.imngs.org) platform using default settings[50]. Removal of primers and technical reads resulted in fragments of approximately 250 bases. Sequencing was performed from both the 3′ and 5′ side resulting in sufficient resolution. IMNGS is a UPARSE based analysis pipeline[51]. Pairing, quality filtering and OTU clustering (97% identity) was done by USEARCH 8.0[52]. The analysis was based on OTUs rather than amplicon sequence variants (ASVs) since we aimed at aggregating taxa at a higher level and wanted to avoid overestimation of prokaryotic diversity due to Intragenomic heterogeneity of 16S rRNA genes[53]. Chimera filtering was performed by UCHIME (with RDP set 15 as a reference database[54]). Taxonomic classification was done by RDP classifier version 2.11 training set 15.8 Sequence alignment was performed by MUSCLE and treeing by Fasttree[55,56]. A total of 21,372,397 V4 reads were generated over two runs. After trimming, quality filtering, removal of potential chimeric reads, de-multiplexing and removal of low abundant operational taxonomic units (OTUs), 15,692,587 sequences belonging to 478 OTUs were retained for downstream analysis. Negative controls were evaluated based on their number of sequences and composition compared to other samples. We used for each batch, sampling blank controls, DNA blank extraction controls and no-template amplification controls, and monitored the lack of contaminant bacterial DNA load herein by a gel-based principle. Moreover, we compared the acquired OTUs composition from the negative controls to our low abundant microbial samples to ensure that our findings were not driven by potential contaminant taxa. Subsequently, samples of the negative controls and with low sequencing depth (less than 6,749 sequences/sample) were excluded from subsequent analysis. For the remaining samples, the number of sequences per sample ranged from 6749 to 239,395 (median 87,227).

**Richness, diversity, taxonomy, and enterotype analyses.** Data normalization, diversity, taxonomical binning and group comparisons were performed using the Rhea package version 1.6[57]. In order to not discard informative data, normalization in Rhea was performed by dividing OTU counts per sample for their total count (sample depth) followed by multiplying all of the obtained relative abundance for the lowest sample depth (6749 reads/sample). Alpha- (observed species and Shannon index) and the generalized Unifrac beta-diversity index were calculated using the Rhea package[58]. Additional beta-diversity indices (weighted Unifrac, unweighted Unifrac, and Bray–Curtis distance) were calculated using the R package (version 3.6.1.) Phyloseq package version 1.30.0[59]. Ordination of samples according to their microbial composition expressed as Hellinger transformed genus abundance data or beta-diversity indices was visualized using Principal Component Analysis (PCA) and Principal Coordinate Analysis (PCoA), respectively. All ordinations were constructed using the R package Phyloseq and included 95% confidence ellipses. Dirichlet multinomial mixtures models (DMM) were used to calculate genus-level enterotypes[60]. When including samples from all time-points, Laplace approximation revealed an optimal number of three clusters.

**Downstream microbial analyses and presentation.** Smoothing of the kinetic for dominant taxa (Fig. 1e and f, as well as Supplementary Fig. 1d and e) was generated using the geom_smooth function of the ggplot package 3.2.1. with default settings. The lines reflect the mean values of the relative abundance. The appearance and disappearance of OTUs of the dominant phyla (Actinobacteria, Proteobacteria, Firmicutes, Bacteroidetes) with age was visualized in Sankey-plots (SankeyMATIC. com). For readability of Sankey-plots, only OTUs present in >10% of all samples per timepoint and with a prevalence of >20% in the entire dataset, were included. Ecosystem specific functional metagenome predictions were created by the novel PICRUSt-iMGMC workflow (using PICRUSt version 1.1.3) with the de novo picked OTUs and using mouse metagenome-assembled genomes linked to 16S rRNA genes[61]. The derived KEGG orthologs were mapped into multiple pathways

or modules. Differentially changed KEGG modules were identified using the pathways enrichment analyses from MicrobiomeAnalyst with default settings[62]. The identification of *lactobacillus*-related OTUs were performed using EZbiocloud and used to construct a phylogentic tree of lactobacilli by MEGA7 (MUSCLE) for alignment and iTOL v4 for the final annotations (itol.embl.de) with default settings.

**Liver metabolomics and bile acid analyses**. The metabolome analyses were carried out with the AbsoluteIDQ® p180 Kit (Biocrates Life Science AG, Austria. The kit allowed identification and quantification of 188 metabolites from 5 compound classes (acyl carnitines, amino acids, glycerophospho-, and sphingolipids, biogenic amines, and hexoses). The kit used flow injection tandem mass spectrometry (FIA) for the non-polar metabolites and LC-MS/MS for the more polar compounds. The integrated MetIDQ Software (version Boron 2623) streamlined the data analysis by automated calculation of metabolite concentrations. Quantification of analytes utilized stable isotope-labeled or chemically homologous internal standards (IS). Controls were included for 3 different concentration levels. For calibration, the kit contained a calibrator mix at 7 different concentrations. The measurements were carried out with an ABI Sciex API5500 Q-TRAP mass spectrometer via Electrospray ionization (ESI) by Multi Reaction Monitoring (MRM) mode for high specifity and sensitivity. 158 MRM pairs were measured in positive ion mode (13 IS) and 2 MRM pairs were measured in negative mode (1 IS). The following additional chemicals for LC-MS were used: water, Millipore; PITC, Fluka; pyridine, Fluka (p.a.); methanol, Merck; Lichrosolv for LC/MS; acetonitrile, Merck; formic acid, Sigma Aldrich. Metabolites were extracted from liver samples by adding $H_2O$/acetonitrile (1:1,v:v) per mg sample followed by homogenization with a tissue disruptor (10 min, 30 Hz, 4 steel balls). The samples were centrifuged (1400×$g$, 2 min) and the supernatant analyzed. The targeted analysis was performed by adding 10 µL of extracted liver sample to the AbsoluteIDQ® p180 Kit (Biocrates Life Science AG, Innsbruck, Austria), following the vendor's instructions[63]. For bile acid measurements the MS-based Bile Acids Kit (Biocrates Life Sciences AG, Innsbruck, Austria), a 96-well plate format assay, was used following the manufacturer's instructions with normalization based on mass[64]. The following settings were used: turbo spray for ion source, 20 for Curtian Gas, medium for CAD Gas, 40 psi for ion source gas 1 and 50 psi for ion source gas 2. For the bile acid measurements we used an ion spray voltage of −4500 V and a temperature of 600 °C; for the p180 kit we used an ion spray voltage of 5500 V and a temperature of 500 °C. All metabolomic data generated in this study have been submitted to the Metabolomics Workbench and have been assigned the Study ID ST001388, ST001389, ST001396, ST001397, the Project ID PR000952 and the Project DOI 10.21228/M8N397.

The contribution of each metabolite to metabolomic variation was derived from age-constrained redundancy analysis (RDA) based on all metabolites stratified in group levels with additional scaling by normalization based on z-scores (Phyloseq package). Moreover, PCA was used to illustrate changes in the composition of the different metabolic groups during the postnatal period for both PC1 and PC2, and PC2 and PC3. For the bile acids the 2nd and 3rd component were chosen for plotting, since the first component was mainly driven by the strong separation observed between the first and subsequent time points (PND1 vs. PND7–56).

**Multi-omics analyses**. Regularized canonical correlation analyses (rCCA) were performed (Mixomics package 6.10.8)[65] to unravel specific correlations between bile acids and OTUs with a minimal presence of 20% in all samples. Samples were excluded from the microbiota-data if they were not measured in the bile acid analyses (i.e. PND1 of litter 1 and 2). Prior to rCCA a hyperbolic sine transformation was used on OTU-counts and a log-transformation for the bile acids. For the estimation of regularization (penalization) parameters λ1 and λ2, the cross-validation (CV) method was used. We used a λ1 = 0.0001, λ2 = 1 with a CV-score = 0.4779644 and 2 components. OTUs with a correlation between −0.3 and 0.3 on the first 2 components were filtered out to optimize the rCCA.

The Spearman's rank correlation coefficient was calculated between bile acids (weight corrected) and bacterial genera (relative abundances) with a minimal presence of 20% in all samples. Benjamini and Hochberg FDR correction was performed to correct for multiple testing ($p > 0.05$). For the heatmap only significant correlations with adjusted $p$-value of <0.05 are shown and sorted according to their PCA loading scores.

**Hepatic gene expression analyses**. Total RNA was isolated from liver tissue using an RNeasy Plus Universal Kit (Qiagen). RNA concentration and purity were assessed by UV absorption measurement, and integrity was checked using agarose gel electrophoresis. 750 ng of total RNA was converted into single-stranded cDNA using a SensiFAST™ cDNA Synthesis Kit (Bioline). cDNA equivalent to ca. 16 ng of total RNA was used as input for real-time PCR analysis, employing Sybr Green chemistry (SensiMix™ SYBR® Hi-ROX Kit, Bioline) and a LightCycler® 480 System (Roche). The following primer pairs were used: *Fgfr4*, 5′-CTGGAGTCTCGGA AGTGCAT-3′ (MMFGFR4*F1) and 5′-GTACACGCGGTCGAACAATG-3′ (MMFGFR4*R1); *β-klotho*, 5′-CACTGTGGGACACAACCTGA-3′ (MMKLB*F1) and 5′-GAGAACTCGGGGATCATGGC-3′ (MMKLB*R1); *Fxr*, 5′-AAGCTTCCA GGGTTTCAGACA-3′ (MMNR1H4*F1) 5′-CTGTGAGCAGAGCGTACTCC-3′ (MMNR1H4*R1); *Abcb11*, 5′-CTATAGCTGCCGCAAAGCAG-3′ (MMABC B11*F1) and 5′-AGCTGCACTGTCTTTTCACT-3′ (MMABCB11*R1); *Abcb4*,

5′-TCTATGACCCCATGGCTGGA-3′ (MMABCB4*F1) and 5′-GTGTTATATT TTTGGGGCAGCG-3′ (MMABCB4*R1); Ntcp, 5′-CCTTGCGCCATAGGGAT CTT-3′ (MMNTCP*F1) and 5′-GTAGCCCATCAGGAAGCCAG-3′ (MMNT CP*R1); *Cyp7a1*, 5′-TCACTGTGCTTCCTGCTTTG-3′ (MMCYP7A1*F2) and 5′-AAGCCATGGGAAGGTATGTG-3′ (MMCYP7A1*R2); *Cyp8b1*, 5′-GGTACG CTTCCTCTATCGCC-3′ (MMCYP8B1*F1) and 5′-GAGGGATGGCGTCTTA TGGG-3′ (MMCYP8B1*R1); *Cyp27a1*, 5′-GACCTCCAGGTGCTGAACAAGA-3′ (MMCYP27A1*F1) and 5′-CTGTTTCAAAGCCTGACGCA-3′ (MMCYP27 A1*R1); Cyp7b1, 5′-TCTCTTTGCCGCCACCTTAC-3′ (MMCYP7B1*F1) and 5′-ATACTTCCCCACAAGGAAGACAG-3′ (MMCYP7B1*R1); *Cyp2c70*, 5′-TGA CCAGGGAGGATGAGTTTTCTG-3′ (MMCYP2C70*F1) and 5′-CACAGGGGGA GCCATTGGT-3′ (MMCYP2C70*R1); Cyp27a1, 5′-GACCTCCAGGTGCTGAAC AAGA-3′ (MMCYP27A1*F1) and 5′-CTGTTTCAAAGCCTGACGCA-3′ (MMC YP27A1*R1). All primer pairs (Sigma-Aldrich) were designed using PrimerBlast, and correct amplicon size was verified by agarose gel electrophoresis. LinRegPCR was used for analysis of real-time qPCR data. Gene expression data was normalized to the geometric mean of two reference genes, viz. *36b4* using the primer pair 5′-TCGTTGGAGTGACATCGTCTT-3′ (MM36B4*F1) and 5′-TCTGCTCCCACA ATGAAGCA-3′ (MM36B4*R1); *Hprt* using the primer pair 5′-CAGTCCCAGC GTCGTGATTA-3′ (MMHPRT*F1) and 5′-TGGCCTCCCATCTCCTTCAT-3′ (MMHPRT*R1) and *Gapdh* using the primer pair 5′-GGGTCCCAGCTTAGGT TCAT-3′ (MMGAPDH*F2) and 5′-CCCAATACGGCCAAATCCGT-3′ (MMGA PDH*R2)[66].

**Absolute abundances (16S rDNA gene qPCR)**. The 16S rDNA gene copy number was determined to quantify the amount of bacterial genomes present in the samples and convert the relative OTU abundances into absolute bacterial numbers. The PCR primers 5′-CCTACGGGNGGCWGCAG-3′ (16S_341_F) and 5′-GACTACHVGGG TATCTAATCC-3′ (16S_805_R) were used[67]. The real-time PCR was performed on a MyiQTM System (BioRad, USA) in a reaction-volume of 25 µl with 12.5 µl iQTM SYBR Green (Biorad), 2 µl template DNA and 0.75 µl primers (10 µM). The cycling conditions were 95 °C for 3 min followed by 35 cycles of 95 °C for 15 s; 55 °C for 20 s and 72 °C for 30 s. Total 16S gene copy numbers were calculated by comparing the CT value to a standard curve with known concentrations of a plasmid encoding the target 16S gene sequences. Absolute OTU numbers were calculated by multiplying the relative abundances with the 16S rDNA gene copy numbers per sample.

**In vitro culture assays**. *Lactobacillus johnsonii* DSM 100219 similar to BA-OTU 4, *Lactobacillus reuteri* DSM 32035 similar to BA-OTUs 7 and 364, *Lactobacillus murinus* DSM 100193 similar to the BA-OTUs 2 and 798 and *Escherichia coli* DSM 28618 previously isolated from the intestine of mice[68] were employed for in vitro growth assays. Lactobacillus and *E. coli* strains were cultured overnight in MRS (Merck, Germany) and BHI (Sigma-Aldrich) medium, respectively. Cultures were diluted 1:100 and incubated in a 96-well plate (at 100 µL/well) in the absence or presence of 1% (w/v) UDCA, TCA, GCA, or βTMCA (Sigma-Aldrich, Taufenkirchen and Biomol, Hamburg) and incubated at 37 °C for 24 h under anaerobic conditions. Bacterial growth was determined by optical density (OD) measurements at 570 nm ($OD_{570}$) and expressed as $OD_{570}$ in medium with 1% bile acid/ $OD_{570}$ in medium with no bile acid supplement.

**Statistical analysis**. A permutational multivariate analysis of variance (PERMANOVA) using distance matrices (vegan package version 2.5.6, vegan::adonis) was performed in each case to determine if the separation of groups was significant. The Kruskal–Wallis or Mann–Whitney U test were used to test for differences with respect to the relative abundance of bacterial phyla, families and genera, as well as for differences in microbial richness and diversity. For these analyses, apart from the default settings we used no cutoff for relative abundance, no cutoff for median abundance, and zero values were included in statistics. In addition, significance of a trend was evaluated to test for differences in alpha-diversity with age, by including it as a continuous variable in a regression model. All statistical tests were performed by two-sided approaches. Data analyses were performed with R (version 3.6.1.) and GraphPad Prism version 8 software. $p$ values are indicated as follows: ****$p <$ 0.0001; ***$p < 0.001$; **$p < 0.01$; and *$p < 0.05$.

**Reporting summary**. Further information on research design is available in the Nature Research Reporting Summary linked to this article.

## Data availability

All data from this study are publicly available. V4 16S rDNA bacterial sequences have been submitted to the Qiita (https://qiita.ucsd.edu/study/description/10719) and ENA (https://www.ebi.ac.uk/ena/data/view/PRJEB33959) databases under accession No. 10719 and ERP116798, respectively. The metbolomic data are available at the NIH Common Fund's National Metabolomics Data Repository (NMDR) website, the Metabolomics Workbench, https://www.metabolomicsworkbench.org/data/DRCCMetadata.php? Mode=Project&ProjectID=PR000952) where it has been assigned the Study ID ST001388, ST001389, ST001396, ST001397 and the Project ID (PR000952). The data can

be accessed directly via it's Project https://doi.org/10.21228/M8N397. This work is supported by NIH grant U2C-DK119886. Source data are provided with this paper.

## Code availability

Only publicly available scripts have been used for the data analysis in this study. Source data are provided with this paper.

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

## Acknowledgements

We would like to thank Xinwei Chang and Martina Bernecker for excellent technical support. This work was supportesd by the priority program SPP1656 (HO2236/9-2 to M.W.H. and BL953/5-2 to A.B. and M.B.), the Collaborative Research Centers CRC1371 (Project-ID 395357507-SFB1371 to A.B. and M.B.) and CRC1382 (Project-ID 403224013—SFB 1382 to M.W.H. and M.v.B.) and the individual grants HO2236/14-1 and HO2236/17-1 (to M.W.H.) from the German Research Foundation (DFG), a grant from the Interdisciplinary Center for Clinical Research (IZKF) within the faculty of Medicine at the RWTH Aachen University (to M.W.H.), as well as a D2 seeding grant from the School of Nutrition and Translational Research in Metabolism (NUTRIM) of Maastricht University (to N.v.B.).

## Author contributions

N.v.B., J.P., and M.W.H. planned the experiments. N.v.B. collected the murine tissues, performed the animal experiments, performed the microbiota analyses, and analyzed the data. F.S and S.W.M.O.D. performed quantitative PCR analyses. U.R.-K., and M.v.B. performed metabolic analyses and analyzed the results. M.B. and A.B. provided organ tissue from germ-free mice. P.H.M.S. provided support with the data analysis. J.P. and M. W.H. supervised the study and N.v.B., J.P., and M.W.H. wrote the manuscript.

## Competing interests

The authors declare no competing interests.
