## [Peer Review File · Nature Communications]

Reviewers' comments:

Reviewer #1 (Remarks to the Author):

This study employed a combination of “omics” approaches to profile gut microbiome, liver metabolites and transcriptomics in mice from birth to 56 days of age, which aimed to identify intrinsic host factors that help shape the gut microbiome during postnatal development. Bio-informatics analysis of the multiple data sets showed a strong correlation of hepatic bile acid composition and gut microbiome, suggesting that bile acids act as host factors driving the postnatal microbiota maturation. Although the experiments are well designed, the study is of descriptive nature with major conclusions reached mainly based on correlations between data sets.

Some of the major concerns are listed below:

1. Fig 3b shows that the abundance of hepatic primary bile acids post-weaning is at least an order of magnitude larger than that of secondary bile acids, therefore the change in the bile acid pool composition was mostly driven by the primary bile acids, which the authors should address on pg 9.
2. Changes in GCA and UDCA in Fig S9 appear to be more related to the initiation of diet, as the data variability does not show significant differences with age. It is difficult to determine changes in TCA since the d7 values are widely divergent. TMCA does not appear to appreciably change after weaning, when comparing between adjacent timepoints. Therefore, there seems to be changes between pre-weaning and post-weaning, but not “age-related” as the authors state.
3. The strong correlation involving UDCA and GCA is puzzling since glycine conjugated bile acids are in trace amount in mice and their effect on gut microbiota is likely minimal at such low concentration.
4. Why were only the small intestine OTUs analyzed with the rCCA analysis? Fig 1 shows the colon has a much richer bacterial representation. That analysis should be carried out as well or rationale for focusing on the small intestine should be added.
5. Why was the oral gavage of bile acids not carried out for 3 days before d7 instead of starting at d7? Since the microbiota begin to change significantly after d7, inoculating between d1 and d7 would seem the best window to impact the baseline bacterial composition.

6. Given the wide spread of the data, the PCoA plots in Fig 4 are not particularly convincing. Perhaps another type of analysis could be used to demonstrate similarity between microbiota at d56 and at d9 following bile acid treatment.

7. Similarly, the data in Figures 4e-f-g have a lot of variability in bacterial abundance, so the *Lactobacillus*/*Escherichia* ratio is not significantly different from adult for any of the conditions.

8. The presentation of the data at the bottom of pg 11 is very confusing as both TMCA and UDCA led to low *Escherichia*.

9. Without metagenomics it is impossible to determine whether the microbiome function is altered by bile acids. Furthermore, the long-term effect of neonatal bile acid treatment was not determined. Therefore, a conclusion that specific bile acids induce development of a microbiome that is “beneficial” cannot be truly determined without knowing how function is altered in the developing microbiome.

10. On pg. 15, the authors state that TMCA administration fostered growth of *Lactobacillus* OTUs containing BSHs and were closely related to *L. reuteri* and *L. jonsonii*. Those relationships were inferred *in silico* and not determined experimentally, which should be noted.

11. How did the authors draw their conclusion that bile acid concentrations and species profiles appear to contribute in environmental filtering or niche-based interactions? Which data showed that?

12. In the discussion, the authors do not delineate limitations of their study. Lack of metagenomics data to determine microbiome function and strain-level correlations with bile acids is a key limitation. Metabolomics measurements only encapsulated a small number of metabolites. It is possible that untargeted metabolomics could reveal other compound classes which play an important role in the development of the microbiome.

Minor Comments:

Pg 5 lines 21-24 Which figure shows that *Mannheimia* at 24 h after birth clusters separately? Is it driving the clustering at 24 hr in Fig S1a? Why would *Corynebacterium* not colonize?

Based on the richness graphs in Figs 1 and S1, it seems as if diversification really begins to change following d7 rather than d1 as stated by the authors. No increase is observed between d1 and d7.

Sankey plots in Fig S3 seem to show the early proteobacteria contribution is most evident in the colon, whereas in the small intestine, early proteobacteria and actinobacteria seem to similarly influence later composition. This differential should be noted.

DMM modeling inset in Fig 1d is difficult to interpret. P1 and P7 datapoints cannot be distinguished. This figure should be bigger and labeled to aid interpretation. In Fig 1d, what are the clusters with the dashed lines referring to vs the solid lines? Are they the different tissues? It is unclear from the graph labeling. Suggest Fig 1d should be in supplement. Fig S3 could be included in Fig 1 instead as it is more informative.

Lines 11 and 12 on page 7 are difficult to understand. The organs should be grouped with the capacity.

Is there a citation supporting the statement that the hepatic metabolome is a proxy for systemic metabolism? The metabolomics analyses performed here were limited to 188 compounds, therefore it is not likely that systemic metabolism could be effectively modeled with these data. This statement should be revised. Furthermore, why was the metabolome not investigated in the feces and/or in blood? There may be other factors contributing to microbiome development that circulate through the leaky gut in early life.

Reviewer #2 (Remarks to the Author):

Van Best et al present a clear report on a study of the development of neonate mice gut microbiota and the concomitant changes in bile acid production and metabolism. The interesting observation comes from the oral administration of some of said bile acids, which causes the microbiota to mature more quickly, partially recapitulating developments that would usually happen later. While the results are not overly surprising, they are for sure interesting and the study is performed using standard tools and without noticeable flaws. The language is clear and the figures informative. I have listed a few instances of missing details in the report and have some minor issues with the presentations, as detailed below.

missing methodological/statistical detail:

- abstract: mention N

- p5/l4-6: mention N; also, what's the difference between 24h and 1 day? from the fact that the profile in 24h looks different to the one on d1, I guess these are different sets of studies, but this is pretty unclear; throughout the manuscript PND1 is referred to as "at birth" - is it the within 24h after birth or on the second day?

- p5/l7-9: the text suggests that 10 time points were studied (10 mice per litter?) but figure 1a suggests 6 time points - which is it?; also the text is missing an indication of numbers of litters and ergo total number of samples

-p8/l20-22: the GF mice were not introduced before, maybe say more about this treatment?

-p10/l25: which adult microbiota samples are these?

-p11/l1-2: is there a statistical basis for the statement, or is this based purely on the impression of the first two PCs?

-p20/l4: add url and/or reference for the platform; what's the rationale for the amplicon length settings? for the used fragments, I'd expect most OTUs to have a length of 294, some taxonomic groups slightly above 300 bases (if the primers were kept- no removal is indicated), so the 200 seems oddly loose and the 300 oddly stringent? what's the rationale for the 97% clustering - as one of the aims of the study was to trace OTUs over time, wouldn't an exact sequence variant approach be more suitable?

-methods:

p25: it's good to analyse absolute bacterial abundances (or approximations thereof) by qPCR. However, relative numbers of reads per OTU from amplicon sequencing are not reflective of real proportions of organisms in a sample (see eg <https://elifesciences.org/articles/46923>). Therefore, this analysis is not valid. As it is not a major point of the manuscript, I'd be not too worried, but would like to see if the conclusions hold if the normalisation to absolute numbers are not performed or if the analysis is performed on presence/absence data or a different normalisation (e.g. as suggested in the linked article).

minor:

-discussion:

general: it's not always clear which statements refer to humans, mice or both

p13/l17-19: high richness in meconium has been observed repeatedly so it's not surprising, e.g. Mueller et al, Genes 2017, Wampach et al, Frontiers in Microbiology 2017, Palmer et al, Plos1 2007 (you also cite this one)

-terminology:

p2/l4: most of the develop is succession rather than evolution

p2/l15: it's unclear what about the maturation is "enhanced" - is it just quicker?

p3/l3: I'd suggest to add "mammalian" or "human" or "murine" before host.

p6/l5: the intestinal content is not a tissue

p7/l21: while I've seen the term in literature, I find "breast milk" inappropriate in mice and would suggest to call it just "milk"; or replace "switch in diet " by "weaning"

p9/l5: If this statement is based on the Picrust analysis, it should read "the predicted abundance of genes for the key enzyme..."

p9/l7: introduce "predicted" before "BSH"

p9/l8: the second part of this sentence "expression of proteins" refers to liver transcriptomes, correct? this should be clarified (at the moment it reads more like either Picrust microbiome genes or liver proteins).

- language:

p3/l9-10: move "human" from before "disease" to before "intestinal"

p4/l17: replace "identical" with "same"

legend Sfig 3 - should be kinetics rather than kinetic, although I am not convinced by the term in this context

p11/l3: replace "enhance" with "increase"

figures:

- Fig 1: what kind of smoothing is applied in figure 1e/f and Sfig 1 d/e? are these mean values? it's unclear how the ellipses for the DMM-clusters are chosen, since they hardly include any points? something's wrong with figure legend "(G) and (G)"

- SFig 3 - why are there these shapes going from nowhere to nowhere around the rectangles on PND1 (e.g Actinobacteria, Proteobacteria)? the legend claims that the height reflects relative number of OTUs - relative to what? the text describes that the plots visualize the increase in diversity, isn't that contradictory to the legend?

Fig 2: y-axis labels are incomplete

Fig 4: a: the pink and orange dots are really difficult to tell apart. I'd suggest adding differentiating symbols, too; d: the y-axis should only go up to 1; the title is redundant with the y-axis label

Fig 5: missing n in Dunn's name; legend should contain more detail regarding the way the tree was built.

Reviewer #3 (Remarks to the Author):

This is a well-designed and performed study, data presented are novel and interesting. Some correlations between bile acid species and hepatic metabolism are speculative and difficult to explain.

Major comment:

Bile salt hydrolase (BSH) deconjugation of bile acids mainly occurs in the colon, not small intestine. Small intestine, mainly in ileum, reabsorbs 95% conjugated bile acids. Also bacterial contents increase and microbiota change from upper intestine to small intestine to colon. Bile acid compositions also varied from indium (mostly conjugated bile acids) to cecum (mostly deconjugated secondary bile acids) to feces (mostly DCA). Correlation of bile acid species and BSH activity in the small intestine to hepatobiliary transport maturation does not make sense and need to be clarified. Overall, this is a very nice study of bile acids and gut microbiome in development from neonate to maturity in mice. Most data are consistent to current understanding of bile acid metabolism in development.

In Fig. 1e, what are the color codes? Same as in 1f? some colors and symbols are difficult to discern in many figures.

Page 8, lines 6-17 need modification. "BSH increased in small intestine"? De-conjugation of bile acids mainly occur in the colon, where bacteria enriched with BSH are located. Most conjugated bile acids are reabsorbed in ileum. "Increased BSH was paralleled by enhanced expression of proteins involved in hepatobiliary transport of bile acids. ...". Conjugated bile acids efflux from liver (Abcb11) and up taken into intestinal (Asbt) in the ileum, thus the increased intestine BSH parallels bile acid transporters is not consistent with enhanced BSH and maturation of the hepatobiliary transport system as stated in these sentences. In the colon, most bile acids are un-conjugated (by BSH) and are transported by passive transport/diffusion to portal circulation or excreted into feces.

Point to point response to referees

Reviewer #1:

This study employed a combination of “omics” approaches to profile gut microbiome, liver metabolites and transcriptomics in mice from birth to 56 days of age, which aimed to identify intrinsic host factors that help shape the gut microbiome during postnatal development. Bio-informatics analysis of the multiple data sets showed a strong correlation of hepatic bile acid composition and gut microbiome, suggesting that bile acids act as host factors driving the postnatal microbiota maturation. Although the experiments are well designed, the study is of descriptive nature with major conclusions reached mainly based on correlations between data sets.

We appreciate the reviewer's comment but would like to remember that we did perform significant interventional *in vivo* challenge experiments (Fig. 4 and 5, Fig. S10). The value of these interventional experiments was acknowledged by reviewer #2 (general comment): “The interesting observation comes from the oral administration of some of said bile acids, which causes the microbiota to mature more quickly, partially recapitulating developments that would usually happen later.”

Major point 1: Fig 3b shows that the abundance of hepatic primary bile acids post-weaning is at least an order of magnitude larger than that of secondary bile acids, therefore the change in the bile acid pool composition was mostly driven by the primary bile acids, which the authors should address on pg 9.

We agree with the reviewer's comment. Since we primarily focused on the small intestine, primary bile acids constitute the major portion of the bile acid pool present even after weaning. This is consistent with the literature and the results of our analysis (Fig. 3, Fig. S8). As suggested, we have now addressed this finding in the manuscript text.

"Age-dependent changes in the concentration of bile acids were noted within the time window of major alterations in the microbiota composition. Since the metabolome analysis was performed on liver tissue to reflect the situation in the small intestine, these changes throughout the postnatal time period primarily reflected the abundance of primary bile acids".

Major point 2: Changes in GCA and UDCA in Fig S9 appear to be more related to the initiation of diet, as the data variability does not show significant differences with age. It is difficult to determine changes in TCA since the d7 values are widely divergent. TMCA does not appear to appreciably change after weaning, when comparing between adjacent time points. Therefore, there seems to be changes between pre-weaning and post-weaning, but not “age-related” as the authors state.

The term “age-related” was used to indicate that changes in the concentration of bile acids occur at different time points during the postnatal and early infant period and thus could explain the observed alterations of the microbiota composition. We agree with the reviewer that major changes in the bile acid concentrations particular of GCA (Fig. S9a) and TCA (Fig. S9d) appear between day 14 and 21 after birth consistent with a potential dietary influence at weaning and have modified the text accordingly. However, other observed changes such as for example the early decrease of α/β TMCA (Fig. S9c) or the late increase of UDCA (Fig. S9b) are most likely not due the change in diet during weaning. To demonstrate this, we have now added the statistical analyses to Figure S9 to illustrate the statistically significant differences between time points.

To address the reviewer's point and indicate that the diet might play a major role but also indicate weaning-independent changes we have modified the text as follows:

"Notably, these patterns are in line with the observed changes in the concentration of specific bile acids between pre- and post-weaning but also early after birth and after weaning (Fig. S9)."

Major point 3: The strong correlation involving UDCA and GCA is puzzling since glycine conjugated bile acids are in trace amount in mice and their effect on gut microbiota is likely minimal at such low concentration.

The reviewer is correct that bile acids in mice are primarily conjugated with taurine. Yet the analysis provides an unbiased approach and we would like to stress that the effect of a specific bile acid on the microbiota might not primarily dependent on its concentration. To address the reviewer's important point, we discuss this issue in the manuscript text.

"Specific direct or indirect effects might explain why the overall effect size of a given bile acid might not primarily depend on the concentration."

Major point 4: Why were only the small intestine OTUs analyzed with the rCCA analysis? Fig 1 shows the colon has a much richer bacterial representation. That analysis should be carried out as well or rationale for focusing on the small intestine should be added.

The decision to focus the rCCA analysis on the small intestinal microbiota was a consequence of the selection of the liver tissue for the metabolome analysis. The liver represents a better proxy for the small intestine as compared to the large intestine (Sayin et al., 2010 and particularly Quinn et al., Nature 2020). For example, the liver metabolome analysis resulted in the identification of conjugated primary bile acids, known to represent the dominant bile acid form in the small intestine, as strongly age-variable metabolites (Fig. 2a-d, Fig. S9).

In addition, the small intestinal microbiota was analyzed since the effect on the small intestine might be clinically relevant. The mucosal immune system is most dominant in the small intestine and microbiota-dependent maturation effects on the small intestinal immune system have been shown to have a great impact on host-microbial homeostasis (e.g. Cahenzli et al., CHM, 2013; Constantinides et al., Science, 2019). This rationale has now been added to the manuscript text.

"We analyzed the hepatic metabolome of the identical animals with the goal to reveal the site-specific spectrum of metabolites. This strategy has previously been shown to reliably reflect the metabolic situation in the small intestine (Sayin et al., 2010; Quinn et al., 2020)."

and

"Conjugated primary bile acids dominate during the immediate postnatal period and also after weaning are present at high concentrations in the small intestine. They may well exert a significant influence particularly on the small intestine where a critical influence of the microbiota composition on the maturation of the mucosal immune system in the neonate has been demonstrated (Cahenzli et al., 2013; Constantinides et al., 2019)."

However, we agree that bile acids in general might of course also influence the colonic microbiota. As requested we therefore extended the rCCA analysis to the colonic microbiota. The results are consistent with our previous results, which might be due to the fact that the concentration of primary bile acids in the neonate is particularly high (Fig. 3b) or that the small intestinal microbiota has a strong influence on the downstream colonic microbiota during this time window. We have added these results in the new Supplemental figure (**new Fig. S10**) and discuss them in the text.

"rCCA analysis of bile acids and microbial OTUs of the colon revealed a similarly high canonical coefficient and dominant influence of TCA, MTCA and UDCA on the colonic microbiota composition (new Fig. S10). This might be due to the high concentration of conjugated bile acids during the"

postnatal period or a strong influence of the small intestinal microbiota on the colonic microbiota during this time window (Fig. 3b)"

Major point 5: Why was the oral gavage of bile acids not carried out for 3 days before d7 instead of starting at d7? Since the microbiota begin to change significantly after d7, inoculating between d1 and d7 would seem the best window to impact the baseline bacterial composition.

Bile acids were administered daily by gavage, a procedure that bears the risk of injury of the vulnerable infant esophagus and subsequent loss of the animal, particularly in very young neonates (< 7 days-old). Also, despite an increase in richness, the microbial community structure in untreated animals is relatively stable between day 7 and 14 (see Fig. 1d). The analysis at day 9 after birth is still well prior to weaning and the dietary change from milk to solid food.

Major point 6: Given the wide spread of the data, the PCoA plots in Fig 4 are not particularly convincing. Perhaps another type of analysis could be used to demonstrate similarity between microbiota at d56 and at d9 following bile acid treatment.

We agree with the reviewer that there is substantial inter-individual variation in the response to bile acid supplementation, which might be related to inter-individual differences in the baseline microbiota (Fig. 1) and/or the dynamic character of the effect of the administered bile acids. Nevertheless, our statistical analysis shows that on average the microbiota of animals supplement with TCA and TMCA becomes significantly more similar to adult mice (Fig. 4d). To better visualize this we have now added box plots visualizing the scores on the first principal component to show the statistically significant separation (included in the **new Fig. S4a and b**).

Major point 7: Similarly, the data in Figures 4e-f-g have a lot of variability in bacterial abundance, so the *Lactobacillus*/*Escherichia* ratio is not significantly different from adult for any of the conditions.

We appreciate the reviewer's comment, which made us realize that we did not accurately state the way we performed the statistical testing. In Fig. 4e-g we tested whether the relative abundance of *Lacobacillus*, *Escherichia* and the ratio between these two genera was significantly different between the control group and the groups supplemented with the respective bile acids. The abundance of these genera in the adult group was added to allow a visual comparison.

Notably, we did not make statistical comparisons between these groups and the adults. The reason for this was because it would require different statistics and hypothesis testing. We performed traditional statistics to proof the hypothesis that bile acid supplementation resulted in an increase of lactobacilli (TMCA and UDCA) and decrease of *Escherichia* (TMCA) when compared to controls. If we would apply the same statistics to compare adults to the different experimental conditions we could only proof which conditions would significantly differ from adults, but not provide information on whether a condition would lead to a significantly more similar composition. This would require equivalence/non-inferiority statistics and including such statistics would likely reduce the clarity of our message.

To address the reviewer's point we now explicitly state in the legend of Fig. 4 that we did not statistically compare the adults with the other groups to clarify this point (see legend of Fig. 4): *"The abundance of these genera (E, F) and the ration of the abundance of Lactobacillus OTUs to Escherichia OTUs (G) in adult mice (Adult) was added to allow visual comparison but was not included in the statistical evaluation."*

Major point 8: The presentation of the data at the bottom of pg 11 is very confusing as both TMCA and UDCA led to low *Escherichia*.

The reviewer is correct that both TMCA and UDCA led to a significantly lower absolute abundance of *Escherichia* (Fig. S11). However, only TMCA but not UDCA also enhanced the abundance of *Lactobacillus* OTUs found in the adult microbiota (OTU 7, 364 and 4). This observation can explain the reduced ability of UDCA to enhance the similarity to the adult microbiota (Fig. 4d). We have modified the paragraph to improve the clarity as follows:

"Although UDCA also decreased the absolute abundance of Escherichia (Fig. S10e), it increased the colonization density of Lactobacillus OTUs not typically found at higher abundance in the adult microbiota (Fig. S10a-c). This finding may explain why UDCA does not enhance the similarity of the neonatal microbial composition to the adult microbiota (Fig. 4d)."

Major point 9: Without metagenomics it is impossible to determine whether the microbiome function is altered by bile acids. Furthermore, the long-term effect of neonatal bile acid treatment was not determined. Therefore, a conclusion that specific bile acids induce development of a microbiome that is "beneficial" cannot be truly determined without knowing how function is altered in the developing microbiome.

The reviewer is correct and we have modified our statement as requested:

"Limitations of our study are the lack of metagenomic data and the restricted pool of metabolites analyzed in hepatic tissue. Metagenomic information would provide additional insight into the functional characteristics of the identified taxa and allow strain-level correlations with detected metabolites. "

Major point 10: On pg. 15, the authors state that TMCA administration fostered growth of *Lactobacillus* OTUs containing BSHs and were closely related to *L. reuteri* and *L. johnsonii*. Those relationships were inferred *in silico* and not determined experimentally, which should be noted.

The reviewer probably refers to line 16, page 15 in the original version of the manuscript text. This sentence describes the rise in *Lactobacillus* OTU7, 364 and 4 after oral TMCA administration (Fig. 5a-c and Fig. S10a-c). We agree that the link to BSH activity was generated *in silico* and the paragraph has been modified to clarify this point :

"βTMCA administration in vivo enhanced the abundance Lactobacillus OTUs that according to available databases contain BSHs and were closely related to dominant murine Lactobacillus species, including L. reuteri (BA_OTU 7 and 364) and L. johnsonii (BA_OTU 4) (Schwab et al., 2014)."

Major point 11: How did the authors draw their conclusion that bile acid concentrations and species profiles appear to contribute in environmental filtering or niche-based interactions? Which data showed that?

Environmental filtering and niche-based interactions were only mentioned in the discussion section as hypotheses to explain the observed effects. We modified the text to clarify this:

"From the perspective of ecological theory, bile acid concentrations and species profiles could potentially promote processes such as environmental filtering or niche-based interactions."

Major point 12: In the discussion, the authors do not delineate limitations of their study. Lack of metagenomics data to determine microbiome function and strain-level correlations with bile acids is a key limitation. Metabolomics measurements only encapsulated a small number of metabolites. It is possible that untargeted metabolomics could reveal other compound classes which play an important role in the development of the microbiome.

As suggested we added a paragraph describing the limitations of our study in particular in respect to the lack of metagenomic data and the restricted set of analyzed metabolites.

"Limitations of our study are the lack of metagenomic data and the restricted pool of metabolites analyzed in hepatic tissue. Metagenomic information would provide additional insight into the function of the identified taxa and allow strain-level correlations with detected metabolites. Also, measurement of a more extended set of metabolites or even an untargeted metabolic approach could reveal other compound classes that play an important role in the development of the early postnatal microbiome. Additional factors that contribute to the microbiome development after birth might be identified by the future analysis of metabolites in other samples such as feces and blood."

Minor comment 1: Pg 5 lines 21-24 Which figure shows that Mannheimia at 24 h after birth clusters separately? Is it driving the clustering at 24 hr in Fig S1a? Why would Corynebacterium not colonize?

We apologize for the misleading comment on the clustering of Mannheimia in this context and have rephrased the sentence and added the transient bloom of Corynebacteria to describe our results on the median abundance of the most dominant taxa during the postnatal period shown in Fig. 1e and f as well as Fig. S1d and e).

"Only the abundance of the genera Corynebacterium and Mannheimia increased temporarily with a peak at 6-18h and 24h or PND1, respectively. (Fig. S1d and e and Fig. 1e and f)."

The reason(s) for these temporary blooms are, to the best of our knowledge, unknown. The changes in the oxygen concentration in the gut lumen after birth, the induction of some kind of antimicrobial host response (Lotz et al., J. Exp. Med. 2006) in the neonatal gut or the temporary access of commensal skin bacteria such as Corynebacteria and Staphylococci to the microbiota might play a role and a sentence has been added to the text to discuss this point:

"Few bacterial genera such as Mannheimia and Corynebacterium exhibited a transient increase in abundance early after birth. The underlying mechanisms are unclear but might involve changes in exposure or altering environmental conditions in the gut lumen such as oxygen concentration, changes in substrate availability and bacterial competition as well as the induction of mucosal host responses. Global bacterial diversification and increasing richness started only thereafter and continued until after weaning in both small intestine and colon."

Minor comment 2: Based on the richness graphs in Figs 1 and S1, it seems as if diversification really begins to change following d7 rather than d1 as stated by the authors. No increase is observed between d1 and d7.

The reviewer is correct that the richness in Fig. S1 decreases during the first 24 hours and that a subsequent increase Fig. 1b-c is only observed after PND7. However, please consider that we do not have values between 24h/PND1 and PND 7 or between PND7 and PND14 and so we do not exactly know at what time point the diversity starts to increase again. The text has been modified to address the reviewer's point.

"Microbial richness increased from PND7 onwards in both small intestine and colon (Fig. 1b and c)."

Minor comment 3: Sankey plots in Fig S3 seem to show the early proteobacteria contribution is most evident in the colon, whereas in the small intestine, early proteobacteria and actinobacteria seem to similarly influence later composition. This differential should be noted.

As suggested we added a sentence to the manuscript text to highlight this difference:

"The early (PND1) Proteobacteria contribution was most evident in the colon, whereas the small intestine was most strongly influenced by proteobacteria and actinobacteria."

Minor comment 4: DMM modeling inset in Fig 1d is difficult to interpret. P1 and P7 datapoints cannot be distinguished. This figure should be bigger and labeled to aid interpretation. In Fig 1d, what are the clusters with the dashed lines referring to vs the solid lines? Are they the different

tissues? It is unclear from the graph labeling. Suggest Fig 1d should be in supplement. Fig S3 could be included in Fig 1 instead as it is more informative.

As suggested, the DMM modeling insert was enlarged and is now displayed as individual supplemental figure also allowing a better discrimination of the PND1 and PND7 data points (**new Fig. S2c**). Additionally, the legend of Fig. 1d was modified to improve the clarity. Solid lines refer to colon samples; dashed lines to small intestinal samples and this information was added to the legend text as follows: “Squares and solid line: colon (C); triangles and dashed line: small intestine (SI).” We appreciate the reviewer’s comment on the Sankey plots (Fig. S3) but we are afraid that by reducing the size and including this quite complex graph into Fig. 1 would reduce the clarity of the figure for the reader. We therefore decided not to move Fig. S3.

Minor comment 5: Lines 11 and 12 on page 7 are difficult to understand. The organs should be grouped with the capacity.

The text has been modified as suggested by the reviewer and now says: “The enzymatic and absorptive capacity of the intestine and the metabolic capacity of the liver mature significantly during the postnatal period (Muncan et al., 2011; Henning 1985; Grijalva and Vakili, 2011).”.

Minor comment 6: Is there a citation supporting the statement that the hepatic metabolome is a proxy for systemic metabolism? The metabolomics analyses performed here were limited to 188 compounds, therefore it is not likely that systemic metabolism could be effectively modeled with these data. This statement should be revised. Furthermore, why was the metabolome not investigated in the feces and/or in blood? There may be other factors contributing to microbiome development that circulate through the leaky gut in early life.

We appreciate the reviewer’s comment and have modified the text in order to enhance the clarity of the text. The liver tissue was analyzed as a proxy for the small intestinal metabolome (rather than the systemic metabolome and this term has been replaced) as previously suggested (Sayin et al., 2010). Notably, a recent study by Quinn et al. (Nature, 2020) clearly demonstrated the close similarity of the metabolome in the liver and small intestine and this reference has been added to justify the choice to analyze the liver tissue to reflect the proximal intestine. Also, the statement has been rephrased as suggested to indicate the restricted panel of metabolites analyzed.

“To allow analysis of the same mice used for the long-term microbiota analysis presented above we investigated a panel of hepatic metabolites as a proxy for the small intestinal metabolome (Quinn et al., 2020).”

As correctly noted, the metabolome of other samples or organs such as feces or blood could be analyzed and this analysis might reveal additional metabolic factors influencing the microbiota. In the present study, however, we focused on liver metabolites and the small intestine microbiota and rather attempted to provide additional supportive functional *in vivo* and *in vitro* results (see Fig. 4 and 5 as well as Fig. S11). To account for the reviewer’s point, we added this important point to the new paragraph on the limitations of the study in the discussion section.

“Also, measurement of a more extended set of metabolites or even an untargeted metabolic approach could reveal other compound classes that play an important role in the development of the early postnatal microbiome. Additional factors that contribute to the microbiome development after birth might be identified by the future analysis of metabolites in other samples such as feces and blood.”

Reviewer #2:

Van Best et al present a clear report on a study of the development of neonate mice gut microbiota and the concomitant changes in bile acid production and metabolism. The interesting observation comes from the oral administration of some of said bile acids, which causes the microbiota to mature more quickly, partially recapitulating developments that would usually happen later. While the results are not overly surprising, they are for sure interesting and the study is performed using standard tools and without noticeable flaws. The language is clear and the figures informative. I have listed a few instances of missing details in the report and have some minor issues with the presentations, as detailed below.

We thank the reviewer for the overall positive judgment and the very thorough and detailed review of our manuscript.

Missing methodological/statistical detail

Major point 1: abstract: mention N

The number of animals per age group for both the short (n=10) and long time kinetic (n=6) is indicated in the experimental scheme shown in Fig. 1a and Fig. S1a as well as the figure legends; the animals used to test the influence of oral bile acid administration is indicated in the legends. As suggested we added this information also to the abstract.

Major point 2: p5/l4-6: mention N; also, what's the difference between 24h and 1 day? from the fact that the profile in 24h looks different to the one on d1, I guess these are different sets of studies, but this is pretty unclear; throughout the manuscript PND1 is referred to as "at birth"- is it the within 24h after birth or on the second day?

The number of animals was added to the text:

"To investigate the postnatal gut microbial development, we obtained the complete small intestinal and colonic tissue from mice aged 0, 6, 12, 18 and 24 h (n=10/age group) as well as 1, 7, 14, 28 and 56 days (n=6/age group) after birth to cover the early developmental stages until adulthood."

The 24h time point was part of a precise time series of 0, 6, 12, 18 and 24 hours after birth (short term kinetic, Fig. S1), generated using the tissues from a single animal per litter for each time point (i.e. n=10 animals per time point = 10 litters to reflect intra-litter variation). The PND1 time point was a part of a second prolonged time series of PND1, 7, 14, 21, 28 and 56 with again a single animal per litter for each time point (long-term kinetic, n=5 animals). Differences between 24h and PND1 can be explained by the fact that PND1 does not represent precisely 24h but rather the first day of life. Thus, although in total ten time points after birth were analyzed they were analyzed in two different sequential studies (short-term and long-term) due to the limited average litter size of C57BL/6 mice. This information has been added to the revised manuscript (material and methods section):

"To monitor microbiota and host metabolic development throughout the neonatal period into adulthood, intestinal and hepatic tissues were obtained from C57BL/6J mice in two different approaches. First, total small intestinal, colon and liver tissues were obtained from C57BL/6J mice aged 1, 7, 14, 21, 28, and 56 days (n=6 per timepoint). In a separate set of experiments, similar tissues were obtained from mice aged 0, 6, 12, 18 and 24 hours (n=10 per timepoint)."

Additionally, a second infographic has been added (**new Fig. S1a**) that together with the improved infographic in Fig. 1a illustrates our approach.

Major point 3: p5/l7-9: the text suggests that 10 time points were studied (10 mice per litter?) but figure 1a suggests 6 time points - which is it?; also the text is missing an indication of numbers of litters and ergo total number of samples.

We analyzed two different groups of animals to generate (i) a precise short-term (0, 6, 12, 18, and 24h) kinetic and (ii) a more extended long-term (PND1, 7, 14, 21, 28, 56) kinetic. For the kinetics, n=10 animals/age group (short-term kinetic) or n=6 animals/per age group (long-term kinetic) were analyzed for each time point. To control for inter-litter variation, tissues for all time points in a given kinetic were taken from animals from one litter, i.e. for the short term kinetic with 5 time points we employed 10 litters with each 5 animals (each one animal for each time point, in total 10 animals per time point); for the long term kinetic with 6 time points we employed 6 litters with each 6 animals (each one animal for each time point, in total 6 animals per time point). This information has been added to the manuscript text, the infographic in Fig. 1a was improved and an additional infographic was added to illustrate the sample collection for the short term kinetic (**new Fig. S1a**).

“To investigate the postnatal gut microbial development, we obtained the complete small intestinal and colonic tissue from mice aged 0, 6, 12, 18 and 24 h (n=10/age group) as well as 1, 7, 14, 28 and 56 days (n=6/age group) after birth to cover the early developmental stages until adulthood. To avoid litter effects, tissues for all time points in a given kinetic were taken consecutively from animals from one litter (Fig. 1a and Fig. S1a – schematic overview of study design).”

Also the number of samples per time point is added in the figure legend for each experiment (see also major point 2, reviewer #2)

Major point 4: p8/l20-22: the GF mice were not introduced before, maybe say more about this treatment?

As suggested we have added more information on GF mice. GF mice are bred continuously under bacteria-free conditions without any further treatment:

“Notably, postnatal upregulation of the key enzyme of the classical bile acid synthesis pathway Cyp7a1 was also observed in germ-free animals. Germ-free animals are bred and raised in the absence of viable bacteria and thus this postnatal upregulation occurred in a developmental, microbiota-independent fashion (Fig. 2h).”

Major point 5: p10/l25: which adult microbiota samples are these?

The adult control samples were obtained and processed at the same as the samples of the neonate mice left untreated or treated with bile acids from an independent group of adult 8-12 week old mice in order to avoid any processing bias. This information has been added in the manuscript:

“Samples from an additional group of adult 8-12 week old animals (n=5) were processed in parallel to provide an adult microbiota control.”

Major point 6: p11/l1-2: is there a statistical basis for the statement, or is this based purely on the impression of the first two PCs?

The statement that "GCA and UDCA had no impact on the microbiota maturity i.e. the similarity to the adult microbiota" is based on the data presentation and statistical analysis shown in Fig. 4d. The distance to the microbial community structure of adult mice (as assessed by the Bray-Curtis dissimilarity index) was statistically significantly smaller in mice supplemented with TMCA or TCA but not GCA or UDCA as compared to non-supplemented mice. The paragraph in the results has been revised to clarify this:

“... administration of TCA and β TMCA significantly increased the richness of the small intestinal microbiota (Fig. 4c) and decreased the distance to the adult microbiota composition based on the Bray-Curtis dissimilarity analysis (Fig. 4d). In contrast, GCA and UDCA had no impact on bacterial richness (Fig. 4c) or microbiota maturity i.e. the similarity to the adult microbiota (Fig. 4d).”

Major point 7: p20/l4: add url and/or reference for the platform; what's the rationale for the amplicon length settings? for the used fragments, I'd expect most OTUs to have a length of 294, some taxonomic groups slightly above 300 bases (if the primers were kept- no removal is indicated), so the 200 seems oddly loose and the 300 oddly stringent? what's the rationale for the 97% clustering - as one of the aims of the study was to trace OTUs over time, wouldn't an exact sequence variant approach be more suitable?

As suggested, the url of the platform (www.imngs.org) has been added to the text. The indicated settings were after removal of primers and technical reads, resulting in fragments of approximately 250 bases and by far most amplicons are within this size range (98.5% with 251-260bp). This means that the length filtering was exactly 50 bp above and below the expected fragment length. Sequencing was performed from both the 3' and 5' side resulting in sufficient resolution. The paragraph describing the approach was extended.

We decided to base the analysis on OTUs rather than ASVs for two reasons. First, we could not follow individual animals longitudinally (since the animals had to be sacrificed to obtain gut and liver tissue) and thus were unable to track specific strains but wanted to aggregate taxa at a higher level. Second, we wanted to avoid overestimation of prokaryotic diversity due to Intragenomic heterogeneity of 16S rRNA genes (Sun et al., Appl. Environ Microbiol. 2013) (see e.g. Fig. 5). This information and the reference have been added to the manuscript text.

"The analysis was based on OTUs rather than amplicon sequence variants (ASVs) since we aimed at aggregating taxa at a higher level and wanted to avoid overestimation of prokaryotic diversity due to Intragenomic heterogeneity of 16S rRNA genes (Sun et al., 2013)."

Methods

Major point 8: p25: it's good to analyse absolute bacterial abundances (or approximations thereof) by qPCR. However, relative numbers of reads per OTU from amplicon sequencing are not reflective of real proportions of organisms in a sample (see eg <https://elifesciences.org/articles/46923>). Therefore, this analysis is not valid. As it is not a major point of the manuscript, I'd be not too worried, but would like to see if the conclusions hold if the normalisation to absolute numbers are not performed or if the analysis is performed on presence/absence data or a different normalisation (e.g. as suggested in the linked article).

We appreciate the reviewer's comment. We are aware of the problems associated with the PCR-based quantification. Nevertheless, we agree with the reviewer that even an approximation of the absolute abundances can provide important and interesting additional information. In the revised manuscript, the data and the statistical analysis of the relative abundances are shown in main figure 5a-e. The data confirm that the described associations are valid without normalization. Nevertheless, as also indicated by the reviewer, we would like to keep the absolute abundance data (new Fig. S11) since we believe that this information might still be of interest for some readers.

Discussion

Minor comment 1: general: it's not always clear which statements refer to humans, mice or both

We carefully went through the discussion section and have modified it to account for this important issue.

Minor comment 2: p13/l17-19: high richness in meconium has been observed repeatedly so it's not surprising, e.g. Mueller et al, Genes 2017, Wampach et al, Frontiers in Microbiology 2017, Palmer et al, Plos1 2007 (you also cite this one)

As suggested we have rephrased the text and added the indicated references (Wampach et al. and Mueller et al.) :

“Consistent with previous reports, monitoring the immediate postnatal time window revealed an initial reduction of microbial richness within the first 24 hours after birth (Palmer et al., 2007; Mueller et al., 2017; Wampach et al., 2017).”

Terminology

Minor comment 3: p2/l4: most of the develop is succession rather than evolution

The text was modified as recommended:

“Following birth, the neonatal intestine is exposed to maternal and environmental bacteria that successively form a dense and highly dynamic intestinal microbiota.”

Minor comment 4: p2/l15: it's unclear what about the maturation is "enhanced" - is it just quicker?

Yes, based on the analysis demonstrated in Fig. 4a and b and the presentation and statistical analysis presented in Fig. 4c and d, the enteric microbiota after TMCA or TCA administration appears to evolve "quicker". The text was modified as recommended using the term "accelerated" :

“Consistently, oral administration of tauro-cholic acid or β -tauro-murocholic acid to newborn mice (n= 7-14 per group) accelerated postnatal microbiota maturation.”

Minor comment 5: p3/l3: I'd suggest to add "mammalian" or "human" or "murine" before host.

The text was modified as recommended :

“All mammalian host body surfaces and in particular the intestinal tract are colonized by a variety of microorganisms.”

Minor comment 6: p6/l5: the intestinal content is not a tissue

The text was modified as recommended:

“While the bacterial richness was similar in both organs at birth (PND1), a significantly higher colonic richness was detected after weaning”.

Minor comment 7: p7/l21: while I've seen the term in literature, I find "breast milk" inappropriate in mice and would suggest to call it just "milk"; or replace "switch in diet " by "weaning"

The text was modified as recommended:

“Amino acids, biogenic amines, acylcarnitines and some glycerophospholipids showed a moderate but significant decrease between these timepoints, most likely due to the switch in diet during weaning (Fig. 2b-d), which is normally completed after day PND21.”

Minor comment 8: p9/l5: If this statement is based on the Picrust analysis, it should read "the predicted abundance of genes for the key enzyme..."

The text was modified as recommended:

“More specifically, the predicted abundance of genes for the key enzyme involved in bacterial bile acid metabolism, the bile salt hydrolase (BSH, KO1442), increased steadily with age reaching the highest small intestinal levels approximately at PND 21-28 (Fig. 3a).”

Minor comment 9: p9/l7: introduce "predicted" before "BSH"

The text was modified as recommended:

“Increased predicted BSH gene abundance was paralleled by enhanced expression of genes encoding proteins involved in the hepatobiliary transport of bile acids ...”

Minor comment 10: p9/l8: the second part of this sentence "expression of proteins" refers to liver transcriptomes, correct? this should be clarified (at the moment it reads more like either Picrust microbiome genes or liver proteins).

The text was modified as recommended:

"Increased predicted BSH gene abundance was paralleled by enhanced expression of genes encoding proteins involved in the hepatobiliary transport of bile acids such as Ntcp (Slc10A1), Abcb11, and Abcb4 as well as proteins involved in the Fxr-Fgf15-Fgfr4 negative feedback loop regulating de novo hepatic bile acid synthesis in total liver tissue as measured by RT-PCR (Fig. S7)."

Language

Minor comment 11: p3/l9-10: move "human" from before "disease" to before "intestinal"

The text was modified as recommended:

"Perturbations of the human intestinal microbiota composition ..."

Minor comment 12: p4/l17: replace "identical" with "same"

The text was modified as recommended:

"Subsequently, we performed a global metabolic screen in the liver tissue of the same animals to provide information ..."

Minor comment 13: legend Sfig 3 - should be kinetics rather than kinetic, although I am not convinced by the term in this context

The text was modified as recommended:

"Sankey plot illustrating the origin and transmission of OTUs between mother and offsprings at the indicated time points after birth (postnatal day, PND)."

Minor comment 14: p11/l3: replace "enhance" with "increase"

The text was modified as recommended:

"Consistently, administration of TCA and β TMCA significantly increased the richness of the small intestinal microbiota (Fig. 4c) and ..."

Figures

Minor comment 15: Fig 1: what kind of smoothing is applied in figure 1e/f and Sfig 1 d/e? are these mean values? it's unclear how the ellipses for the DMM-clusters are chosen, since they hardly include any points? something's wrong with figure legend "(G) and (G)"

As correctly indicated by the reviewer, smoothing of the kinetic in Fig. 1e/f and Fig. S1d/e reflects the mean values of the relative abundance. To generate these figures, the geom_smooth function of the ggplot package using default settings was employed as described in the material & methods section:

"Smoothing of the kinetic for dominant taxa (Fig. 1e and f as well as Fig. S1d and e) was generated using the geom_smooth function of the ggplot package with default settings. The lines reflect the mean values of the relative abundance."

The ellipses of the DMM clusters depict the 95% confidence interval and this information has been added to the legend text:

"Three DMM-clustering profiles confirming the distinct microbial community structures of small intestine and colon PND1-14 (C1, green symbols), small intestine PND21-56 (C2, orange symbols) and colon PND21-56 (C3, blue symbols) ($p < 0.001$, Permanova). The ellipses depict the 95% confidence

interval."

The legend text was corrected. It now says "E and F" instead of "G and G".

Minor comment 16: - (a) SFig 3 - why are there these shapes going from nowhere to nowhere around the rectangles on PND1 (e.g Actinobacteria, Proteobacteria)? (b) the legend claims that the height reflects relative number of OTUs - relative to what? (c) the text describes that the plots visualize the increase in diversity, isn't that contradictory to the legend?

(a) The shapes on PND1 have been removed from the graph as requested (Fig. S3).

(b) The rectangle height reflects the relative number of OTUs within a given phylum at that specific time-point. This means that we expressed the number of OTUs observed at a specific time-point relative to the total number of OTUs for that phylum as observed within the entire dataset.

We clarified this in the legend text (legend Fig. S3):

"The rectangle height indicates the relative number of OTUs (number of observed OTUs at a given time-point divided by the total number of OTUs within a phylum across all time-points) and the rectangle color reflects the age of the animals."

The manuscript text was modified accordingly and now says:

"The presence, origin and loss of OTUs in small intestinal and colon tissue were visualized in Sankey plots for all postnatal time points as well as for the maternal source".

(c) We modified the text to account for the reviewers comment. The text now says:

"This analysis illustrated the contribution of OTUs from all four major phyla to the microbial ecosystem at the different time points after birth"

Minor comment 17: Fig 2: y-axis labels are incomplete

The y-axis labels in Fig. 2f, g, and h were modified as recommended:

"relative mRNA expression".

Minor comment 18: Fig 4: a: the pink and orange dots are really difficult to tell apart. I'd suggest adding differentiating symbols, too; d: the y-axis should only go up to 1; the title is redundant with the y-axis label

As suggested we modified Fig. 4a to allow a better discrimination of the symbols by adding differentiating symbols. The y-axis of Fig. 4d was changed as recommended (maximal value 1); the title of Fig. 4d was removed (Fig. 4).

Minor comment 19: Fig 5: missing n in Dunn's name; legend should contain more detail regarding the way the tree was built.

The name of the test (Dunn's) was corrected:

"Dunn's post-test and correction for multiple comparisons".

More information regarding the way the phylogenetic tree was built was added in the legend of Fig. 5f.

"The phylogentic tree was constructed by MEGA7 (MUSCLE) for alignment and iTOL v4 for the final annotations."

More information is provided in the material and methods section:

"Taxonomic classification was done by RDP classifier version 2.11 training set 15.8 Sequence alignment was performed by MUSCLE and treeing by Fasttree (Edgar et al., 2004; Price et al., 2010)."

Reviewer #3:

This is a well-designed and performed study, data presented are novel and interesting. Some

correlations between bile acid species and hepatic metabolism are speculative and difficult to explain.

We appreciate the reviewer's general comment and completely agree that more research needs to be done to fully explain the observed findings (see paragraph on limitations of the study in the discussion section).

Major comment 1: Bile salt hydrolase (BSH) deconjugation of bile acids mainly occurs in the colon, not small intestine. Small intestine, mainly in ileum, reabsorbs 95% conjugated bile acids. Also bacterial contents increase and microbiota change from upper intestine to small intestine to colon. Bile acid compositions also varied from indium (mostly conjugated bile acids) to cecum (mostly deconjugated secondary bile acids) to feces (mostly DCA). Correlation of bile acid species and BSH activity in the small intestine to hepatobiliary transport maturation does not make sense and need to be clarified. Overall, this is a very nice study of bile acids and gut microbiome in development from neonate to maturity in mice. Most data are consistent to current understanding of bile acid metabolism in development.

We appreciate the reviewer's general comment on the quality of our study. We analyzed the expression of the hepatobiliary transport molecules and regulatory circuit for the bile acid synthesis not to allow a direct correlation with specific bile acids but to demonstrate that dynamic changes occur during the postnatal period. We believe that these findings support our hypothesis that bile acids influence the postnatal establishment of the small intestinal microbiota. We completely agree with the reviewer that almost all (95%) conjugated bile acids are reabsorbed in the small intestine and that the majority of BSH activity is found in the colon. However, for example many lactobacillus species, prominent commensal bacteria of the small intestine, exhibit BSH activity and the presence of deconjugated bile acids has been reported in the small intestine (Ridlon et al., J Lipid Res, 2006; Sayin et al., Cell Metab., 2013; Ridlon et al., Gut Microbes, 2016). Although this might represent a minor fraction of the total BSH activity and lead to only low concentrations of deconjugated bile acids in the small intestine, it would still enhance the spectrum of bile acids and might thereby exert a significant influence on the low-density microbiome of the small intestine.

Major comment 2: In Fig. 1e, what are the color codes? Same as in 1f? some colors and symbols are difficult to discern in many figures.

The reviewer is correct the legend in Fig. 1f is also valid for Fig. 1e and the legend was modified to clarify this. The symbols were modified in Fig. 4a and b to allow a better discrimination.

Major comment 3a: Page 8, lines 6-17 need modification. "BSH increased in small intestine"? Deconjugation of bile acids mainly occur in the colon, where bacteria enriched with BSH are located. Most conjugated bile acids are reabsorbed in ileum.

We believe that this is a misunderstanding. We completely agree with the reviewer that commensal bacteria of the colon harbor strong BSH activity and this strong colonic BSH activity causes the vast majority of bile acid deconjugation and subsequent generation of secondary bile acids in the body. However, here we are interested in the influence of bile acids mainly in the small intestine and also the small intestinal microbiota includes some bacterial members that encode BSH activity. These bacteria might be affected by the presence of their substrates. For example, lactobacilli are a dominant group of small intestinal commensals (see Fig. 1) and many of them carry BSH activity (DiMarzio et al., PLoS One, 2017; O'Flaherty et al., mSphere, 2018). Importantly, although we observe a growth benefit *in vitro* and an increase in abundance *in vivo* of certain Lactobacillus OTUs that harbor predicted BSH activity and thus discuss a potential influence of bile acid deconjugation on the

small intestinal microbiota, this of course does not mean that we are questioning that the vast majority of bile acid deconjugation occurs in the colon.

To address the reviewer's comment, we have modified the text:

" More specifically, the predicted abundance of genes for the key enzyme involved in bacterial bile acid metabolism, the bile salt hydrolase (BSH, KO1442), increased steadily with age reaching the highest small intestinal levels approximately at PND 21-28 (Fig. 3a). Notably, much higher abundances of BSH activity harboring bacterial taxa are observed in the large intestine "

Major comment 3b: "Increased BSH was paralleled by enhanced expression of proteins involved in hepatobiliary transport of bile acids. ...". Conjugated bile acids efflux from liver (Abcb11) and up taken into intestinal (Asbt) in the ileum, thus the increased intestine BSH parallels bile acid transporters is not consistent with enhanced BSH and maturation of the hepatobiliary transport system as stated in these sentences. In the colon, most bile acids are un-conjugated (by BSH) and are transported by passive transport/diffusion to portal circulation or excreted into feces.

Please remember that these sentences refer to hepatic gene expression (Fig. S7), bile acid analysis in liver tissue (Fig. 3b) and the small intestinal microbiota (Fig. 3a). For reasons delineated in the response to major point 4, reviewer #1, we in this study mainly focused on the small intestine (and not the colon). Here we wanted to highlight the fact that age-dependent changes are observed for (i) the predicted BSH gene abundance (Fig. 3a) and (ii) the expression of host genes involved in the hepatobiliary transport of bile acids and the Fxr-Fgf15-Fgfr4 negative regulatory axis (Fig. S7). We believe that these results support our hypothesis that major changes in the composition and local concentration of bile acids in the intestinal lumen may occur and that these may thus contribute to the observed changes in the composition of the enteric microbiota. However, we agree with the reviewer that the observed kinetic for the predicted BSH activity and the gene expression are different (predicted BSH activity peaks at PND21-PND28 whereas the bile transport and FXR regulatory axis gene expression rises continuously until PND56) and do not parallel. We have therefore modified the paragraph as follows:

"In addition, increased expression of genes encoding proteins involved in the hepatobiliary transport of bile acids such as Ntcp (Slc10A1), Abcb11, and Abcb4 as well as proteins involved in the Fxr-Fgf15-Fgfr4 negative feedback loop regulating de novo hepatic bile acid synthesis was observed after birth in total liver tissue as measured by RT-PCR (Fig. S7)."

REVIEWERS' COMMENTS:

Reviewer #1 (Remarks to the Author):

The authors have extensively revised the paper by adding bacteria data in the colon, and addressing the major criticisms. My major concerns that limit my enthusiasm for an otherwise valid result are the results are focused on BA metabolism and microbial changes mainly in the small intestine. The authors rationalize this as it impacts the immune system, but this isn't a focus of the paper. Additionally, the correlation involving UDCA and GCA is puzzling since glycine conjugated bile acids are in trace amount in mice and their effect on gut microbiota and bile acid reabsorption in the colon is likely minimal at such low concentration.

The observed kinetics for the predicted BSH activity and the gene expression are different (predicted BSH activity peaks at PND21-PND28 whereas the bile transport and FXR regulatory axis gene expression rises continuously until PND56) and do not parallel.

If one accepts these limitations the paper is much improved.

Jacob Friedman

Reviewer #2 (Remarks to the Author):

Van Best et al present a revised description of the development of neonate mice gut microbiota and the changes in bile acid production during the same period, which also includes an experimental oral administration of bile acids that caused microbiota to mature more quickly. I would like to thank the authors for the clarifications in their revised manual. I have only a few points that I feel the need to insist on.

Former Major point 4: I am sorry that I was unclear in my previous comment. Your methods section does not provide detail on the number of samples, the breed of mice, or their relationships and I'd like to see this omission mended.

Former Major point 7: Thank you for the clarifications on the methods. I don't completely buy your rationale for the use of OTUs, however. A) The results presented in Figure S3 and lines 141-144 would clearly benefit from the highest possible resolution. B) Overestimation of microbial diversity is a problem that is more severe for some 97% OTU clustering methods than the exact sequencing variants methods (e.g. 10.1371/journal.pone.0227434, 10.7717/peerj.5364). However, since you've picked the OTU clustering method with the lowest introduction of spurious OTUs, I would not want to make you redo the whole sequence analysis and pretty figures for a minor detail. Maybe you can adjust the wording in the relevant section and methods description?

I also appreciate the additions made in response to reviewer 1's questions and wish you good luck with the publication.

Reviewer #3 (Remarks to the Author):

Authors answered all my comments satisfactorily and made a diligent effort in revising this manuscript.

Reviewer #1

The authors have extensively revised the paper by adding bacteria data in the colon, and addressing the major criticisms. My major concerns that limit my enthusiasm for an otherwise valid result are the results are focused on BA metabolism and microbial changes mainly in the small intestine. The authors rationalize this as it impacts the immune system, but this isn't a focus of the paper. Additionally, the correlation involving UDCA and GCA is puzzling since glycine conjugated bile acids are in trace amount in mice and their effect on gut microbiota and bile acid reabsorption in the colon is likely minimal at such low concentration.

The observed kinetics for the predicted BSH activity and the gene expression are different (predicted BSH activity peaks at PND21-PND28 whereas the bile transport and FXR regulatory axis gene expression rises continuously until PND56) and do not parallel.

If one accepts these limitations the paper is much improved.

Jacob Friedman

We thank the reviewer for her/his constructive criticism and fair and overall positive judgment. We have added his remark on the lack of colon data to the limitations of the study paragraph (page 19, line 4-5) and modified our statement on the bile transport and FXR regulatory axis gene expression as suggested (page 9, line 16-20).

Reviewer #2

Van Best et al present a revised description of the development of neonate mice gut microbiota and the changes in bile acid production during the same period, which also includes an experimental oral administration of bile acids that caused microbiota to mature more quickly. I would like to thank the authors for the clarifications in their revised manual. I have only a few points that I feel the need to insist on.

1. Former Major point 4: I am sorry that I was unclear in my previous comment. Your methods section does not provide detail on the number of samples, the breed of mice, or their relationships and I'd like to see this omission mended.

The number of examined animals is illustrated in the experimental outline in Fig. 1a and Supplementary Figure 1a and the number of analyzed samples is indicated in the legend text for each experiment and. A summary of the total number of animals used in this study has been added to improve the clarity (see page 20, line 22-25). Mice were bred locally as indicated on page 20, line 7-8.

2. Former Major point 7: Thank you for the clarifications on the methods. I don't completely buy your rationale for the use of OTUs, however. A) The results presented in Figure S3 and lines 141-144 would clearly benefit from the highest possible resolution. B) Overestimation of microbial diversity is a problem that is more severe for some 97% OTU clustering methods than the exact sequencing variants methods (e.g. 10.1371/journal.pone.0227434, 10.7717/peerj.5364). However, since you've picked the OTU clustering method with the lowest introduction of spurious OTUs, I would not want to make you redo the whole sequence analysis and pretty figures for a minor detail. Maybe you can adjust the wording in the relevant section and methods description?

I also appreciate the additions made in response to reviewer 1's questions and wish you good luck with the publication.

We thank the reviewer for his constructive criticism and understanding. We believe that there are good arguments for both, to use either OTUs or ASVs. We decided to use OTUs and therefore listed arguments to support this decision (page 23, line 18-21). We are not sure how to adjust the wording here as suggested.

Reviewer #3

Authors answered all my comments satisfactorily and made a diligent effort in revising this manuscript.

We thank the reviewer for his positive judgment.